# TradeFM: A Generative Foundation Model for Trade-flow and Market Microstructure

## Abstract

Learning generalizable representations from the high-frequency, heterogeneous event streams of financial markets is a significant challenge. We introduce TradeFM, a foundation model that learns the universal dynamics of market microstructure. Pre-trained on billions of equities transactions, TradeFM uses a novel scale-invariant feature representation and a universal tokenization scheme to form a unified representation, enabling generalization without asset-specific calibration. We validate the quality of the learned representations by demonstrating that model-generated rollouts in a closed-loop simulator successfully reproduce canonical stylized facts of financial returns. We robustly evaluate the model's ability to generalize to temporally and geographically out of sample data, as well as its ability to match real distributions of quantities like log returns and spreads. TradeFM provides a high-fidelity engine for synthetic data generation and downstream agent-based modeling.

## 1 Introduction

Financial markets are complex systems characterized by high-frequency, non-stationary, endogenous dynamics, driven by interactions of participants (Bouchaud, 2010). The fundamental driver of this process is order/trade-flow, the sequence of buy and sell orders submitted to the market (Sirignano & Cont, 2021). Modeling this process is a formidable challenge due to the heterogeneity of market participants, the asynchronous nature of transaction data, and the dramatic shifts in statistical properties across different assets and time periods (Pasca, 2015).

While traditional approaches often build asset-specific models, there is strong evidence for universal features in price formation that generalize across diverse markets. Sirignano & Cont (2021) demonstrated that a single deep learning model trained on pooled data from a diverse set of stocks can significantly outperform asset-specific models. This provides the core motivation for our work: to build a **foundation model** that learns generalizable representations of market mechanics directly from raw, multi-asset order flow data.

We leverage the Transformer architecture, whose success in capturing long-range dependencies has been proven by Large Language Models (LLMs) that learn general-purpose representations from vast, diverse datasets (Vaswani et al., 2017; Bommasani et al., 2021). By treating the stream of multi-featured trade events as a structured sequence, we apply these powerful sequence modeling techniques to the financial domain.

The contributions of this paper are fourfold:

1. **TradeFM**: We introduce TradeFM, a large-scale, decoder-only generative Transformer for market microstructure, pre-trained on billions of transactions from a diverse set of equities to learn a unified representation of trade-flow dynamics.

2. **Learning from Partial Observations**: A core design principle of our work is learning from a partially observed market state. This reflects the realistic, incomplete information available to any single market participant and enhances the model's practical applicability.

3. **Scale-Invariant Representation and Tokenization**: We present an end-to-end methodology for processing raw, high-frequency data, including a scale-invariant feature repre-

sentation and a lightweight universal tokenization strategy that enables a single model to generalize across diverse assets and liquidity regimes without asset-specific calibration.

4. **Closed-Loop Market Simulation**: We integrate the pre-trained TradeFM with a deterministic market simulator, creating a high-fidelity, closed-loop environment for generating realistic market rollouts, studying second-order effects like market impact, interactive fine-tuning, and training learning-based agents.

## 2 BACKGROUND

### 2.1 THE MECHANICS OF MODERN ELECTRONIC MARKETS

To provide context for a general AI/ML audience, we briefly introduce the core concepts of market microstructure fundamental to this work, which are standard features of modern electronic markets (Hasbrouck, 2007).

Financial markets are predominantly organized around a **Limit Order Book (LOB)**, a real-time record of all outstanding orders for a security that functions as a continuous, double-sided auction. It consists of a **bid** (buy) side and an **ask** (sell) side; the midpoint between highest bid and the lowest ask is an asset's **midprice**. The ease with which an asset can be bought or sold quickly at a stable price is the asset's **liquidity**.

Market participants interact with the LOB through a sequence of actions, collectively known as **order flow**. Participants may submit **limit orders** with a specific price limit, which sit on the book waiting to be matched. The distance between the **order price** and the midprice is the **price depth**, quoted in **ticks** (the minimum price increment, typically $0.01) or **basis points** (0.01% of the price). They may also submit **market orders** for immediate execution against resting limit orders starting at the best bid/ask, and **cancellations** to withdraw resting orders. When an incoming order is matched with a resting one, a **fill** (or trade execution) occurs. This matching process is generally governed by a deterministic **price-time priority** algorithm, where orders are first prioritized by price and then by time of submission. These elements and mechanisms constitute **market microstructure**.

### 2.2 STYLIZED FACTS AS EMERGENT PROPERTIES

The strategic interactions of market participants give rise to endogenous market dynamics (Bouchaud, 2010). These dynamics, in turn, give rise to universal and persistent statistical properties known as **stylized facts**. These facts are observed across a wide range of assets, markets, and time periods, and serve as a crucial benchmark for the realism of any generative market model (Cont, 2001; Ratliff-Crain et al., 2023). Key stylized facts include:

- **Heavy-Tailed Returns**: The distribution of price returns is leptokurtic (heavy-tailed). This means that extreme price movements occur far more frequently than would be predicted by a Gaussian distribution, a critical consideration for risk management.

- **Volatility Clustering**: Price volatility is not constant. Periods of high volatility tend to be followed by more high volatility, and periods of calm tend to be followed by calm. This is observed as a positive and slowly decaying autocorrelation in measures of volatility, such as squared or absolute returns.

- **Lack of Autocorrelation in Returns**: Consistent with the efficient market hypothesis, asset prices are considered to follow a random walk. Consequently, the autocorrelation of asset returns is statistically insignificant beyond very short time lags.

## 3 RELATED WORK

The modeling of market microstructure has evolved from explicit, theory-driven formulations toward implicit, data-driven representation learning. Our work continues this trajectory, positioning a generative foundation model as the natural next step to learn universal market dynamics directly from raw, heterogeneous data.

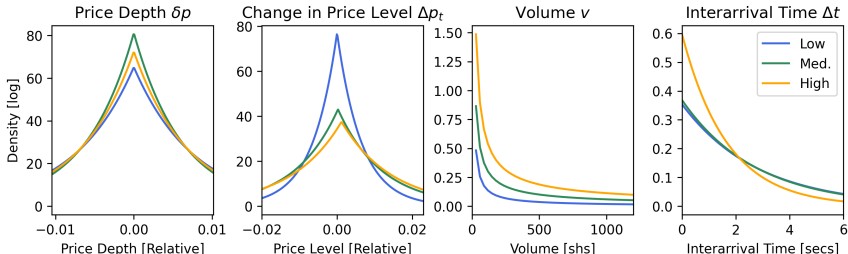

Figure 1: Canonical distributions for core trade features, conditioned on liquidity. Price features are leptokurtic (Laplace); volume follows a heavy-tailed power-law; and interarrival time is exponential.

### 3.1 MARKET MICROSTRUCTURE AND ORDER-FLOW MODELING

**Classical Stochastic Models**  A significant body of literature models order arrival times using point processes, such as Hawkes processes, to capture the self-exciting nature of order flow (Bacry et al., 2015). More sophisticated approaches like Compound Hawkes processes have also been proposed, which combine Hawkes-processes to model interarrival times with other fitted empirical distributions to model additional features like volumes and price depths (Jain et al., 2024). While providing strong theoretical grounding, these models rely on specific parametric assumptions (e.g., Gaussianity) that are unable to capture the heavy-tailed nature of market returns.

**Agent-Based Models**  Agent-based models simulate market dynamics by defining the behavior of individual participants and observing the emergent properties of the system (Byrd et al., 2019). While ABMs have historically required hand-crafting agent behaviors, recent approaches have shown success in calibrating agents on real market data (Dwarakanath et al., 2024). Our work contributes to this line of research by enabling the learning of complex market dynamics, which can serve as a foundation for more sophisticated agent-based modeling.

**Early Deep Learning Models**  The application of deep learning to LOB data was pioneered by models like DeepLOB (Zhang et al., 2019). These models demonstrated the potential of learning features directly from data but were typically trained on a subset of instruments. This limits their ability to learn universal representations across diverse assets and market conditions.

### 3.2 TRANSFORMERS AND FOUNDATION MODELS IN FINANCE

The success of the Transformer architecture in capturing long-range dependencies has led to its widespread application in domains ranging from genomics (Ji et al., 2021), to time-series forecasting (Wen et al., 2022), to non-trading areas of finance, such as modeling payment transactions (Raman et al., 2024). More recently, domain-specific foundation models have emerged. A prominent example is MaRS, a market simulator with a generative foundation model backbone (Li et al., 2024). While our work builds on many of the design principles established in Li et al. (2024)'s comprehensive framework, TradeFM distinguishes itself in two critical dimensions. First, its pre-training dataset is explicitly constructed to **maximize heterogeneity**, covering thousands of assets across multiple sectors and a wide spectrum of liquidity regimes. This is essential for learning truly universal market representations. Second, and more fundamentally, TradeFM addresses **cross-asset generalization** at the feature level, engineering features to ensure that the model learns representations that are directly comparable across all assets.

## 4 PROBLEM FORMULATION

We formulate the task of modeling market microstructure as a generative, autoregressive sequence modeling problem. Let the market dynamics be represented by a sequence of discrete trade events, $E = (e_1, e_2, \ldots, e_T)$. The objective is to learn the conditional probability distribution $P(e_t|e_{<t})$, where $e_{<t}$ denotes the sequence of all events preceding $e_t$. By learning this distribution, the model can generate realistic sequences of future trade events, effectively simulating the market's evolution.

## 4.1 TRADE EVENT REPRESENTATION

A single trade event $e_t$ is a multi-feature tuple capturing the state of the market at the moment of a transaction. Formally, an event is represented as $e_t = (\Delta t_t, \delta p_t, v_t, a_t, s_t)$, where the core features are: $\Delta t_t$: Interarrival time since the previous event (seconds); $\delta p_t$: Price depth of the transaction (basis points); $v_t$: Volume of the transaction (shares); $a_t$: The action/order type (e.g., limit, cancellation); $s_t$: The side of the initiating order (buy/bid or sell/ask). The distributions of these features are depicted in Fig. 1.

## 4.2 KEY TECHNICAL CHALLENGES

Modeling this data stream presents several key challenges inherent to high-frequency markets: the **Heterogeneity and Distribution Shift** across thousands of assets and varying time periods; the **Sparsity and Irregularity** of the asynchronous event stream; the **Partial Observability** of the true market state from transaction data; and a **High-Dimensional, Multi-Modal Feature Space** combining continuous and categorical values.

# 5 DATA AND FEATURE ENGINEERING

Our methodology is designed to process raw, heterogeneous transaction data at scale and transform it into a standardized format suitable for a generative foundation model. This pipeline consists of data curation, robust feature engineering, and a novel tokenization scheme.

## 5.1 DATA SOURCES AND SCALE

The model is pre-trained on a proprietary dataset built from billions of raw, tick-level US equities transactions, spanning 368 trading days from February 2024 to September 2025, across 9,172 unique assets. This represents over 19 billion tokens across 1.9 million date-asset pairs. We employ a temporal hold-out strategy, reserving January 2025 onward across all assets for the test set, yielding a training set of 10.7 billion tokens and a test set of 8.7 billion tokens. The tokenizer is calibrated on the first 30 days of the training data, February 2024. For evaluating out-of-distribution generalization we also hold out one month of data from APAC regions, namely Jan. 2025, for both Japan and China.

## 5.2 MID-PRICE ESTIMATION

A robust estimate of the true market mid-price ($p_t^{\text{mid}}$) is critical for normalizing price-related features. In our partial-information setting, we estimate this from the observed stream of transaction execution prices ($p_t^{\text{exec}}$). Naive approaches like simple rolling average of execution prices are insufficient, as a fixed-width window (e.g., 50 trades) is not comparable across assets with different liquidity levels. Time-based windows (e.g., 2 seconds) can help, but still fail to account for trade volume.

The conventional solution is the Volume-Weighted Average Price (VWAP) (Berkowitz et al., 1988). To make this estimator more reactive to recent information, we introduce **Exponentially-Weighted Volume-Weighted Average Price (EW-VWAP)**. This is calculated by maintaining separate exponential moving averages (EMAs) for the volume-weighted price and the volume itself. The EW-VWAP at time $t$ is then the ratio of these two values: $\hat{p}_t^{\text{EW-VWAP}} = \text{EMA}(p_t^{\text{exec}} \cdot v_t)/\text{EMA}(v_t)$. This serves as our mid-price estimate $\hat{p}_t^{\text{mid}}$.

The smoothing factor $\alpha$ is determined by a time-based halflife, ensuring that the estimate gives more weight to larger and more recent trades, providing a stable and representative price benchmark. Further details are in Appendix: Section A.3 and Figure 9.

## 5.3 SCALE-INVARIANT FEATURE CONSTRUCTION

The statistical properties of trading data vary widely across sectors, liquidity profiles, and nominal prices. In raw dollar, share, and second terms, price depths, volumes, and interarrival times for an asset like AAPL may differ greatly from those of a penny stock. Trade representations must therefore be carefully designed to enforce homogeneity across assets. (Sirignano & Cont, 2021) posits that with such proper representation, mechanics of price formation are universal and invariant.

To address the challenge of heterogeneity, we construct a set of scale-invariant features from the raw event data. While price-related features are computed as unit-less ratios, we often refer to them in terms of basis points (bps) for interpretability, where a ratio of 0.01 corresponds to 100 bps.

- **Interarrival Time** ($\Delta t_t$): Wall clock time since the previous event: $w_t - w_{t-1}$, in seconds.

- **Log-Transformed Volume** ($v_t$): To compress the wide dynamic range of order sizes, which follow heavy-tailed, power-law distributions (Vyetrenko et al., 2019), we apply a logarithmic transformation: $v_t = \log(1 + V_t)$ where $V_t$ is the raw share volume.

- **Normalized Price Depth** ($d_t$): The depth of a limit order with order price $p_t^{\text{order}}$, relative to the mid-price: $d_t = \frac{p_t^{\text{order}} - \hat{p}_t^{\text{mid}}}{\hat{p}_t^{\text{mid}}}$. This representation is comparable across differently priced assets, unlike prior work using price depths in ticks.

- **Relative Price Level vs. Open** ($\Delta p_t$): To capture intraday market movement, we measure the current mid-price relative to the day's opening price ($p_0$): $\Delta p_t = \frac{p_t^{\text{mid}} - p_0}{p_0}$.

Fig. 8, Appendix A demonstrates distributional stationarity of relative features (vs tick based). Fig. 14, Appendix D.3 demonstrates the subsequent temporal stability of scale-invariant features.

## 5.4 Data Composition: Market and Participant-Level Sequences

In downstream applications, TradeFM can be used to model the behavior of the entire market (e.g., for synthetic data generation) or that of individual participants (e.g., for agent-based modeling). To support this flexibility, our training data includes sequences aggregated at both the market level and the participant level, with approximately a 1.6:1 market-to-participant ratio at the token level. We provide the model with a binary indicator feature, $I_{MP}$, to distinguish between these two contexts.

## 6 Tokenization

Standard Transformer architectures, as applied in natural language processing, operate on univariate sequences where each element is a single token from a discrete vocabulary. Our trade event data is a sequence of multi-feature tuples, each comprising a mix of continuous and categorical values. The core challenge of tokenization is to map this event stream into a univariate discrete sequence.

### 6.1 Binning Strategy and Outlier Handling

We discretize each continuous feature by partitioning its distribution into a fixed number of bins. For price-related features, which have symmetric but highly peaked distributions, we employ **Equal-Frequency Binning** (quantile-based). This ensures that the bins in the dense central region of the distribution have a higher resolution, while still capturing the less frequent values in the tails. For log-transformed features, like volume and interarrival time, we use **Equal-Width Binning**. Applying equal-width bins to the logarithmic values creates bins that are effectively logarithmic in the original feature space, providing a way to represent values that span multiple orders of magnitude.

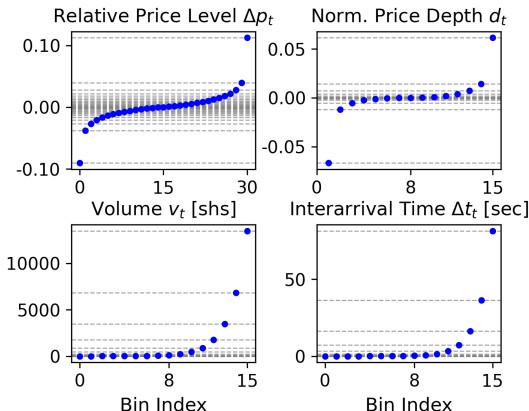

Figure 2: Calibrated bin edges for our hybrid tokenization scheme. Price features (top) use quantile-based binning for high resolution near the mean; volume and time (bottom) use logarithmic bins to capture their wide dynamic range.

This hybrid approach ensures a relatively uniform token distribution, preventing the model from wasting capacity on rare tokens or nearly-empty bins. Before binning, we exclude outliers above the 99th percentile for each feature, and additionally exclude outliers below the 1st percentile for price depth and price level. We reserve

special bins to represent these out-of-range values to prevent the model from allocating excessive capacity to extremely rare events. We calibrate this tokenizer on the first 30 days of our data.

## 6.2 MULTI-FEATURE TOKEN COMPOSITION

While our model's input at each time step is multi-featured, the decoder is trained to predict a single, unidimensional token, representing the core trade event. Thus, we combine the discrete bin indices of the trade-related features, $(i_{\Delta t}, i_{\delta p}, i_v, i_a, i_s)$, into a single composite integer, $i_{\text{trade}}$.

This is accomplished by treating the set of feature indices as digits in a mixed radix (or mixed base) number system. Each feature's bin index is a "digit," and the number of possible values for the subsequent features acts as the "base" at each position. For a concrete example of the encoding process, see Sec. A.4.

We use the following number of bins for each feature: $n_{\delta p} = 16$ for price depth, $n_v = 16$ for volume, $n_{\Delta t} = 16$ for interarrival time, $n_s = 2$ (buy or sell) for side, and $n_a = 2$ (add or cancel order) for action type. The composite trade token $i_{\text{trade}}$, a single integer encoding all constituent features, is calculated as:

$$i_{\text{trade}} = (i_a \times n_s \times n_{\delta p} \times n_v \times n_{\Delta t}) + (i_s \times n_{\delta p} \times n_v \times n_{\Delta t}) + \\ (i_{\delta p} \times n_v \times n_{\Delta t}) + (i_v \times n_{\Delta t}) + i_{\Delta t} \tag{1}$$

This yields a vocabulary size of 16,384 for the predictable trade tokens. The model input at each time step is a tuple containing this composite token along with several non-predicted features used for conditioning. These contextual features are provided as separate inputs, and are not part of the composite trade token $i_{\text{trade}}$ calculated in Eq. 1. The contextual features are:

- $i_l$: The liquidity bin index ($n_l = 3$), determined by binning each asset into low, medium, or high liquidity ranges based on its Average Daily Volume (ADV).
- $i_{\Delta p_t}$: The price level change bin index ($n_{\Delta p} = 32$).
- $I_{MP}$: A binary indicator for market-level vs. participant-level sequences.

The final input is $[i_{\text{trade}}, i_l, i_{\Delta p_t}, I_{MP}]$. This formulation allows the model to be conditioned on the broader market context while focusing its predictive power on the next trade event.

## 7 TRADEFM ARCHITECTURE

**TradeFM is a decoder-only Transformer**, trained from scratch with a custom configuration. The architecture is based on the Llama family and incorporates enhancements including grouped-query-attention (GQA) and rotary positional encoding (RoPE) (Touvron et al., 2023). Our model size is **524 million parameters**, a size chosen based on Chinchilla scaling laws (Hoffmann et al., 2022) for our dataset size (see Appendix Section B.2 for detailed hyperparameter choices). The model is trained on **3 Nvidia A100 GPUs**; we include detailed training setup details in Appendix Section B.3.

### 7.1 TABULAR INPUT EMBEDDING

We employ a tabular embedding approach to handle our multi-feature input tokens (as described in Section 6.2). Each feature in the input tuple $[i_l, I_{MP}, i_{\Delta p_t}, i_{\text{trade}}]$ is first projected into its own embedding space using an embedding table. These embedding vectors are concatenated and passed through a linear projection layer to create a unified representation in the model's hidden dimension.

## 8 MARKET SIMULATOR

To evaluate the realism of our generative model, we require an environment that can simulate the market execution a sequence of predicted trades. We build a lightweight, deterministic simulator tailored to our specific experimental setting. Our simulator serves two critical functions: 1) it provides the dynamic, state-dependent price level features required by our model during generative rollouts, and 2) it allows us to test if the model's generated trade flow can reproduce the well-known stylized facts of asset returns.

## 8.1 DETERMINISTIC DESIGN

The simulator is designed to mimic the core mechanics of a modern electronic exchange. It maintains a limit order book (LOB) for an asset, an internal clock, and an estimate of the market mid-price (the midpoint of the best bid and ask). The order matching engine employs **price-time priority**: incoming orders are matched with the best-priced order on the opposite side of the book, with ties in price broken by selecting the earliest-submitted resting order (Nasdaq Equity Trading Rules). Before using the simulator to validate our large trade model, **we validate the realism of the simulator itself via stylized facts** discussed in the Appendix, Section D.2.

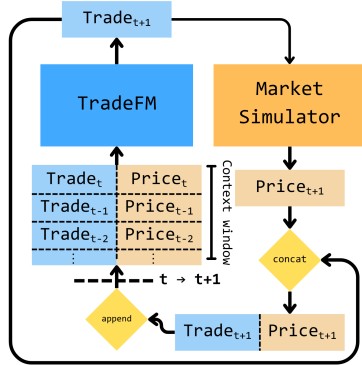

Figure 3: The closed-loop simulation architecture. TradeFM predicts a trade, the Market Simulator executes it, and the updated market state is fed back to the model.

## 8.2 THE CLOSED-LOOP ROLLOUT

The simulator creates a closed-loop system where the model and the environment interact dynamically. This process, which we term a *rollout*, is shown in Fig. 3 and proceeds as follows:

1. **Prediction**: Given a history of market events, TradeFM predicts the next event token, $i_{\text{trade}}$. We use multinomial sampling with a repetition penalty of 1.2 to decode the token.
2. **Execution**: The predicted event (e.g., a new order or cancellation) is passed to the simulator, which executes it against the LOB according to its price-time priority rules.
3. **State Update**: The simulator updates its internal state, including the LOB and the mid-price.
4. **Feedback**: The market state is used to generate the contextual features for the next time step, which are appended to the history and fed back into TradeFM to generate the next prediction.

This recursive loop allows us to generate long, dynamic sequences of market activity. Crucially, it enables the study of second-order effects like price impact, as the model's own predictions influence the market state that conditions its future predictions.

## 9 EXPERIMENTS

Evaluating generative models of financial markets presents a unique challenge due to the non-stationary nature of the data. The underlying data-generating process of a market can shift over time, meaning that simple predictive accuracy (e.g., next-token perplexity) on a static test set may not be a reliable measure of a model's true capabilities. Additionally, prices follow random walks, so there exists no single "correct" midprice trajectory.

We adopt a more robust evaluation framework by assessing the model's ability to reproduce the invariant stylized facts of market behavior. These statistical properties are emergent features of endogenous market dynamics (Section 2.2). A model that can generate synthetic data exhibiting these facts has not merely memorized a historical pattern, but has learned the underlying, time-invariant "grammar" of the market.

We conduct three experiments to validate our approach. First, we evaluate the model's realism to reproduce **key stylized facts** of asset returns. Second, we perform a rigorous quantitative evaluation of the **distributional fidelity of the generated order flow**. Third, we test the **model's ability to generalize** across different market conditions and be controlled by its conditioning features.

**Baselines** We compare TradeFM against a **calibrated Zero-Intelligence (ZI) agent** baseline (Gode & Sunder, 1993; Farmer et al., 2005) to test whether TradeFM learns complex, conditional market dynamics or merely reproduces the market's basic structural properties. The ZI agent's orders are drawn from empirical distributions of key features, and it interacts with the same market simulator in an identical evaluation pipeline. Additionally, we include a **Compound Hawkes** baseline (Bacry et al., 2015; Jain et al., 2024), which models the arrivals of trade events separated by side and action, and their corresponding volumes and price depths. More details are provided in Appendix B.6.

| $\Delta t_r$ | Kolmogorov-Smirnov | | | Wasserstein | | |
|---|---|---|---|---|---|---|
| | ZI | Hawkes | TradeFM | ZI | Hawkes | TradeFM |
| 10 | 0.198 | 0.295 | **0.064** | 0.003 | 0.002 | **0.001** |
| 30 | 0.255 | 0.288 | **0.092** | 0.007 | 0.005 | **0.002** |
| 60 | 0.302 | 0.262 | **0.122** | 0.012 | 0.009 | **0.004** |
| 120 | 0.346 | 0.173 | **0.145** | 0.021 | 0.015 | **0.008** |

Table 1: Distances of log return distributions from real, for all methods, across return intervals $\Delta t_r$ in seconds. We report **Kolmogorov-Smirnov** and **Wasserstein** distances.

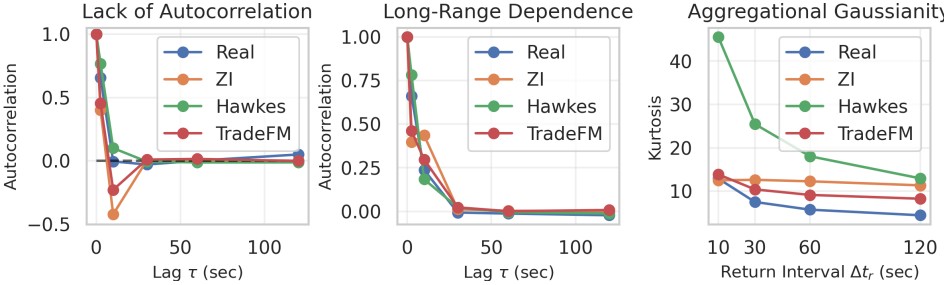

Figure 4: **TradeFM Model Validation** against canonical stylized facts. Simulated returns exhibit: (Left) near-zero autocorrelation, (Middle) slowly decaying autocorrelation of absolute returns (volatility clustering), and (Right) heavy tails and aggregational Gaussianity.

## 9.1 EXPERIMENT 1: STYLIZED FACT REPRODUCTION

To validate the realism of the generated market trajectories, we evaluate their ability to reproduce key stylized facts of log returns ($r_{t,\Delta t}$). We generate 10 rollouts of 1,024 events for 9 assets across 3 liquidity tiers, for each of 9 held-out months, conditioned on a context of 1,024 real historical events. We compute autocorrelations over time lags $\tau$, and evaluate kurtosis over return intervals $\Delta t_r$.

**Results** As summarized in Figure 4 and Table 1, our simulations successfully reproduce the target stylized facts, demonstrating a close correspondence with real market data. Specifically, we observe:

- **Lack of Autocorrelation**: The autocorrelation of simulated log returns (left panel) quickly decays to statistically insignificant levels as the lag $\tau$ increases. This is consistent with the efficient market hypothesis. The ZI baseline exhibits spurious positive autocorrelation.

- **Long-Range Dependence**: The autocorrelation of absolute log returns (middle panel) decays slowly, indicating that our model has captured the long-memory nature of volatility clustering.

- **Heavy Tails**: The kurtosis of simulated returns (right panel) is high for short time scales ($\Delta t_r$), confirming the presence of heavy tails. TradeFM significantly outperforms the baselines in capturing the leptokurtic nature of returns.

- **Aggregational Gaussianity**: As $\Delta t_r$ increases, the kurtosis of TradeFM correctly approaches that of a normal distribution, capturing the reversion towards normality over longer time horizons.

Table 1 provides a quantitative evaluation of the same – the Wasserstein distance ($W_1$) and Kolmogorov-Smirnov (K-S) statistic between real and generated log return distributions from each approach. Our method outperforms all baselines in both metrics, across return intervals.

## 9.2 EXPERIMENT 2: QUANTITATIVE FIDELITY

While reproducing stylized facts confirms that the model captures emergent market dynamics, it does not assess how well the generated order flow aligns with reality. We conduct a quantitative evaluation of distributional fidelity, adopting frameworks established in recent benchmarks for generative order

| Quantity | Kolmogorov-Smirnov | | | Wasserstein | | |
|---|---|---|---|---|---|---|
| | ZI | Hawkes | TradeFM | ZI | Hawkes | TradeFM |
| Spreads | 0.400 | 0.218 | **0.212** | 0.375 | **0.302** | 0.367 |
| IA Times | 0.651 | 0.515 | **0.236** | 0.415 | 0.626 | **0.385** |
| OPD | 0.436 | 0.281 | **0.174** | 0.390 | 0.348 | **0.279** |
| OBI | 0.237 | 0.155 | **0.142** | 0.200 | 0.165 | **0.113** |
| Bid Vol. | 0.460 | **0.296** | 0.371 | 0.616 | 0.278 | **0.143** |
| Ask Vol. | 0.391 | 0.380 | **0.327** | 0.638 | 0.198 | **0.146** |

Table 2: Mean statistics across months for each quantity – interarrival (IA) times, order price depths (OPD), orderbook imbalance at the best bid / ask price (OBI), bid / ask volume – and method.

flow models (Nagy et al., 2025). As in the benchmark we mean-variance normalize distributions before computing Wasserstein distance to make this metric comparable between quantities.

Using the rollouts described in Section 9.1, we compute the Kolmogorov-Smirnov statistic and Wasserstein distance between real and generated distributions for key microstructure variables, including Order Volume, Interarrival Times, Bid-Ask Spreads, and Orderbook Imbalance. This evaluation averages results over 9 assets over 3 liquidity tiers over 9 held-out months. As shown in Table 2, TradeFM achieves consistently lower distance metrics than baseline approaches, demonstrating superior fidelity in reproducing the statistical properties of market data. We provide detailed results in Appendix D.3, Figure 16.

### 9.3 EXPERIMENT 3: GENERALIZATION AND CONTROLLABILITY

To validate our claim that TradeFM learn a universal grammar of market microstructure that generalizes beyond the assets and time periods seen during training, we perform extensive out-of-distribution (OOD) evaluations.

**Geographic Zero-Shot Generalization** We evaluate the model, trained exclusively on US equities, on a hold-out set of assets from APAC markets (China and Japan). We detail these held-out datasets in Appendix D Table 5. Figure 5 shows the distribution of perplexity scores for one month of data (held out for all geographies), with significant overlap between US and APAC. The minimal degradation in perplexity on unseen markets confirms TradeFM's generalization capabilities.

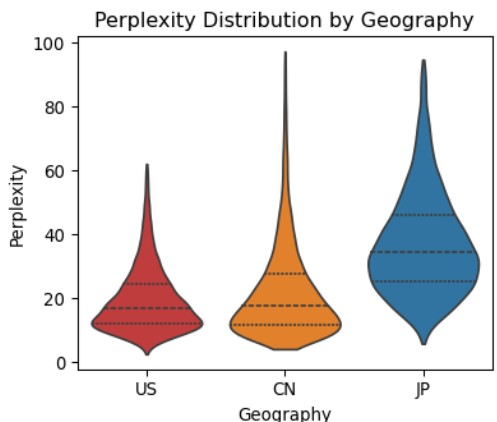

Figure 5: Distributions of perplexities across geographies.

**Temporal Robustness** Financial markets are non-stationary, with regimes shifting over time. We evaluate model performance over an extended hold-out period from Jan - Sep 2025, exhibiting heightened volatility, distinct from the 2024 training set. As detailed in Appendix D.3, Figure 16, distance metrics remain stable over this 9-month horizon.

**Controllability** Finally, we verify that the model respects its conditioning tokens and that its output can be reliably steered via the indicator features ($I_{MP}$ and $i_l$). We generate 256 context-free trajectories of 512 tokens each, for every combination of market-participant and liquidity indicators. We then analyze the statistical properties of the raw generated orders by computing the standard deviation of volumes and interarrival times for each condition.

Figure 6 shows that modulating the indicator tokens has a significant and intuitive effect on the statistical properties of the generated orders with two trends:

- The variance of both volume and interarrival time is consistently higher for market-level generation than for participant-level, aligned with the intuition that the aggregate behavior of an entire market is inherently more volatile than the behavior of a single participant.

- The model also captures linear relationships between liquidity and the variance of interarrival times and volumes.

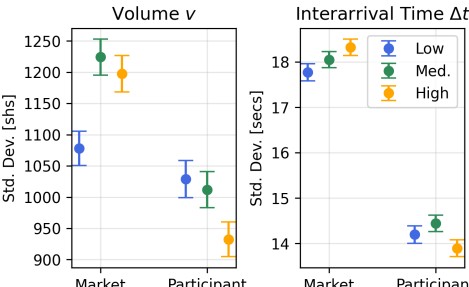

Collectively, these results demonstrate that TradeFM has learned a generalizable, conditional model of market behavior, capable of generating statistically and contextually appropriate order flow.

Figure 6: Standard deviation of generated volumes and interarrival times, conditioned on liquidity ($i_l$) and observation level ($I_{MP}$). The model produces distinct order flow, demonstrating controllable generation.

## 10 APPLICATIONS AND EXTENSIONS

The successful validation of TradeFM as a high-fidelity generative model of market microstructure opens up several avenues for future research and practical application. We focus here on synthetic data generation but discuss market simulation, stress testing, and multi-agent modeling in the Appendix (Sections D.5 and D.4). The learned latent space representations of trading data can also be frozen and used as embeddings for downstream tasks such as forecasting or trade classification.

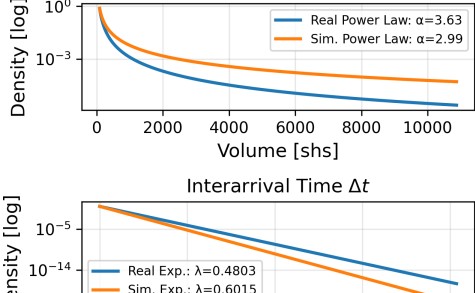

### 10.1 SYNTHETIC DATA GENERATION

TradeFM can serve as a powerful engine for generating high-fidelity, privacy-preserving synthetic financial data. The realism of the generated data is validated not only at the level of emergent price dynamics (Figure 4), but also at the granular level of individual orders. As shown in Figure 7, the distributions of simulated order volumes and interarrival times closely match the power-law and exponential distributions observed in real data, respectively. This high-fidelity generation is valuable for several reasons:

Figure 7: Validation of order-level statistics. (Top) Simulated order volumes closely follow the power-law observed in real data. (Bottom) Simulated interarrival times match the exponential distribution of real data.

- **Backtesting Trading Strategies**: Synthetic data allows for the robust testing of trading algorithms against a wide range of plausible market scenarios, reducing the risk of overfitting to a single historical path (Potluru et al., 2024).

- **Augmenting Sparse Datasets**: For illiquid assets where historical data is sparse, the model can be used to generate additional data to facilitate more robust analysis and model training.

- **Sharing Privacy-Preserving Data**: The model can generate realistic datasets for academic or public use without revealing sensitive, proprietary transaction information.

## 11 CONCLUSION

We have shown that the complex, emergent dynamics of financial markets can be learned directly from raw, heterogeneous order flow. Our end-to-end methodology, which combines a novel scale-invariant feature representation with a universal tokenization scheme, allows a single generative Transformer to generalize across thousands of diverse assets without asset-specific calibration.

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

# A APPENDIX

## A.1 FOUNDATION MODELS FOR STRUCTURED DATA

This work draws inspiration from the success of foundation models beyond NLP, in domains with structured sequential data. This paradigm has been successfully adapted by treating domain-specific sequences as a form of "language." In genomics, for example, models treat DNA or protein sequences as sentences to learn fundamental biological patterns (Ji et al., 2021). For general-purpose time-series forecasting, large pre-trained models have demonstrated strong zero-shot performance on unseen series (Garza et al., 2023; Zhou et al., 2021). Similarly, for tabular data, Transformers pretrained on a diverse collection of tables can perform inference on new, smaller tables without task-specific fine-tuning (Badaro et al., 2023; Hollmann et al., 2025). In finance, by framing order flow as a structured language of market events, our approach aligns with this proven paradigm, arguing for its direct applicability to the unique challenges of financial data.

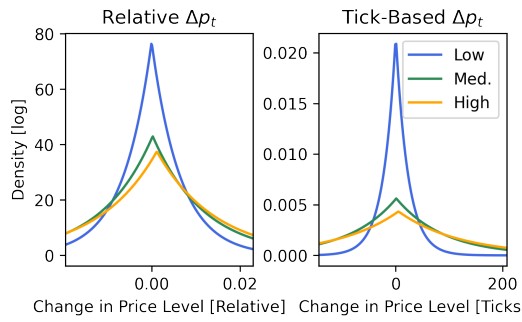

Figure 8: Properties of tick-based vs. relative feature construction for the sample feature $\Delta p_t$, across liquidity profiles. We find that relative features generalize better across assets than absolute, tick-based features.

## A.2 THE TRANSFORMER AS A NATURAL FIT

The Transformer architecture is uniquely suited to address these challenges. Its core components map naturally to the fundamental properties of order flow data:

- **Self-Attention**: The attention mechanism is designed to capture complex, long-range dependencies within a sequence. This makes it an ideal tool for modeling the long memory and intricate, non-linear interactions inherent in order flow.

- **Sequence-to-Sequence Framework**: As an autoregressive, sequence-based model, the Transformer inherently handles the asynchronous, event-driven nature of the data, where the time between events is itself a feature to be learned.

- **Adapting to Multi-feature Sequences**: While Transformers excel at processing univariate text, our trade events are multi-feature tuples. A key challenge is thus to effectively discretize and tokenize these mixed-type features into a processable sequence, which motivates our novel tokenization and embedding methodology.

We develop an end-to-end methodology to build TradeFM, a generative foundation model for market microstructure. The following four sections detail each component of our pipeline: our data processing and scale-invariant feature engineering (Section 5), our universal tokenization scheme (Section 6), the TradeFM model architecture (Section 7), and the closed-loop market simulator used for evaluation (Section 8).

## A.3 MID-PRICE ESTIMATION

A robust estimate of the true market mid-price ($p_t^{\mathrm{mid}}$) is critical for normalizing price-related features. While dedicated market data sources for this exist, they are often expensive. Given our access to raw transaction data, we seek to estimate this value directly. In our partial-information setting, we primarily observe the execution prices ($p_t^{\mathrm{exec}}$) for consummated trades. The raw stream of $p_t^{\mathrm{exec}}$ is a noisy version of the true mid-price $p_t^{\mathrm{mid}}$.

A naive approach, such as a simple rolling average of execution prices, is insufficient. A fixed-width window of trades is not comparable across assets with different liquidity levels; a 50-trade window may span less than a second for a highly liquid asset but several hours for an illiquid one. A time-based window (e.g., 2 seconds) is more relevant, but a simple average still fails to account for trade

| Time-step | Time-stamp | Asset | Avg. Daily Vol. (shs) | Midprice ($) | Action | Side | Order Price ($) | Vol. (shs) |
|---|---|---|---|---|---|---|---|---|
| | | | | $\vdots$ | | | | |
| 42 | 09:45:30 | AAPL | 53,496,022 | 182.45 | ADD | BUY | 182.44 | 500 |
| 43 | 09:45:38 | AAPL | 53,496,022 | 182.48 | ADD | SELL | 182.50 | 750 |
| 44 | 09:45:52 | AAPL | 53,496,022 | 182.50 | CANCEL | BUY | 182.49 | 300 |
| | | | | $\vdots$ | | | | |

Table 3: Toy example of trading activity for an imaginary AAPL trade sequence, demonstrating the multifeature and heterogeneous nature of our data pre-tokenization.

volume. For example, an average that gives equal weight to a 1,000-share trade at $10.00 and a 1-share trade at $9.00 would produce a misleading estimate.

The conventional solution to this problem is the volume-weighted average price (VWAP), which is

$$\hat{p}_t^{\text{VWAP}} = \frac{\sum_{i=0}^{W} v_{t-i} p_{t-i}^{\text{exec}}}{\sum_{i=0}^{W} v_{t-i}} \tag{2}$$

To make this estimator more reactive to recent information, we introduce **Exponentially-Weighted Volume-Weighted Average Price (EW-VWAP)**. This is calculated by maintaining two separate exponential moving averages (EMAs): one for the volume-weighted price and one for the volume itself. For each incoming trade with execution price $e_t$ and volume $v_t$, we update the EMAs for the numerator ($N_t$) and denominator ($D_t$) as follows:

$$N_t = \alpha \cdot (p_t^{\text{exec}} \cdot v_t) + (1 - \alpha) \cdot N_{t-1}$$
$$D_t = \alpha \cdot v_t + (1 - \alpha) \cdot D_{t-1}$$

The EW-VWAP at time $t$ is then the ratio of these two values:

$$\hat{p}_t^{\text{EW-VWAP}} = \frac{N_t}{D_t} \tag{3}$$

The smoothing factor $\alpha$ is determined by a time-based halflife, ensuring that the estimate gives more weight to larger and more recent trades in a temporally consistent manner. This provides a stable and representative price benchmark that reflects the price at which the bulk of recent market activity has occurred.

A.4   TOKENIZATION EXAMPLE

Given the imaginary sequence of trade events $e_t$ constructed in Table 3, our features for timestep $t = 43$ are as follows:

- $\Delta t_t = w_t - w_{t-1} = 09\text{:}45\text{:}38 - 09\text{:}45\text{:}30 = 8\text{sec}$
- $\delta p_t = \frac{o_t - p_t}{p_t} = \frac{\$182.50 - \$182.48}{\$182.48} = \frac{\$0.02}{\$182.48} = 0.011\% = +1.1\text{bps}$
- $v_t = 750\text{shs}$
- $a_t = \text{Add Order}$
- $s_t = \text{Sell}$

Using our calibrated bins, we would discretize these features to the bin indices:

- $i_{\Delta t_t} = \text{bin } 11$
- $i_{\delta p_t} = \text{bin } 7$
- $i_{v_t} = \text{bin } 7$
- $i_{a_t} = \text{bin } 0$

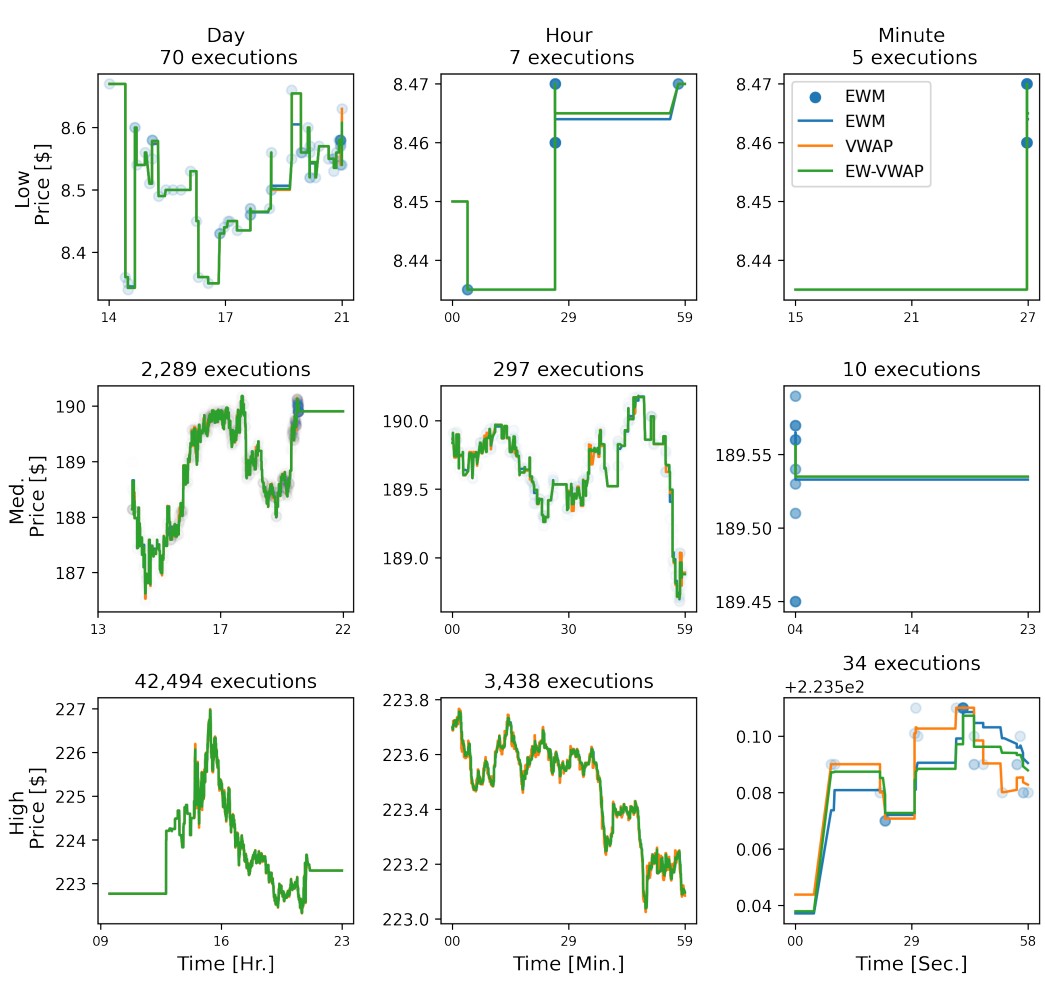

Figure 9: Comparison of mid-price estimators across time scales and liquidity levels. Our proposed EW-VWAP provides a more stable and responsive estimate than standard VWAP or EWM, closely tracking the executed fill prices across different time scales and liquidity regimes.

- $i_{s_t} = \text{bin } 1$

Using Eq. 1 would yield:

$$i_{trade} = 6,011$$

Assuming an opening price $p_0 = \$179.50$, we would have change in price level feature:

$$\Delta p_t = \frac{p_t - p_0}{p_0} = \frac{\$182.50 - \$179.50}{\$179.50} = 0.017\% = +1.7\text{bps}$$

Discretizing this using our bins would give $i_{\Delta p_t} = 19$. Based on the asset's average daily volume of 53 million, which falls into the high liquidity bin, our liquidity indicator $i_l$ would be 2. If this sequence was a market-level sequence, we would have market-participant indicator $I_{MP} = 0$.

Our final model input would then be:

$$[i_l, I_{MP}, i_{\Delta p_t}, i_{\text{trade}}] = [2, 0, 19, 6011].$$

## B    REPRODUCIBILITY GUIDE

### B.1    MODEL BACKBONE

TradeFM is a decoder-only Transformer, trained from scratch with a custom configuration. The architecture is based on the Llama family (Touvron et al., 2023) and incorporates modern enhancements for efficiency and performance, including:

- **Grouped-Query Attention (GQA)**: Balances the performance of Multi-Head Attention with the reduced memory bandwidth of Multi-Query Attention, enabling faster inference and larger batch sizes.
- **Rotary Position Embeddings (RoPE)**: Encodes relative positional information by applying a rotation to query and key vectors, which has been shown to improve generalization for long sequences.

### B.2    MODEL HYPERPARAMETERS AND SCALING

The model size is guided by the Chinchilla scaling laws, which suggest a compute-optimal ratio of approximately 20 training tokens per model parameter (Kaplan et al., 2020; Hoffmann et al., 2022). Given our dataset of 10.7 billion tokens, this implies a target model size of around 525 million parameters. Our final hyperparameters are as follows:

- **Context Length**: 1,024 tokens
- **Hidden Layers**: 32
- **Embedding Dimension**: 1,024
- **Intermediate MLP Size**: 4,096
- **Attention Heads**: 32
- **Key-Value Heads (GQA)**: 8 heads, 4 groups
- **Total Parameters**: 524.4 Million

The 1:4 ratio between the embedding dimension and the intermediate MLP size is chosen in accordance with best practices for Transformer models (Petty et al., 2024).

### B.3    TRAINING CONFIGURATION

We train the model on an AWS instance with 3 Nvidia A100 GPUs, each with 80GB of RAM. All training is performed in `fp16` half-precision. To achieve an effective batch size of 4,032, we use a per-device batch size of 24 and gradient accumulation over 56 steps. For further memory

| Model Size | Num. Train Tokens | Batch Size | GPUs | Train Time / Epoch (hrs) | Optimization |
|---|---|---|---|---|---|
| 125M | 2.6B | 24 | 3xA100 | 17 | Accelerate |
| 250M | 6.4B | 32 | 4xA10G | 29 | DeepSpeed |
| 500M | 10.7B | 24 | 3xA100 | 125 | Accelerate |

Table 4: Training configuration for different model sizes.

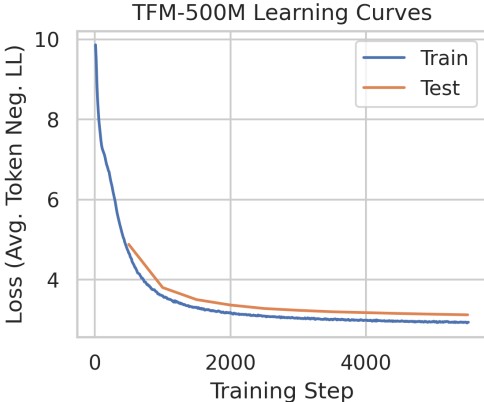

Figure 10: Learning curve for TradeFM-500M, reporting the average token negative log likelihood over the test set constituting Jan. through Sept. 2025 as loss.

optimization and training acceleration, we use the Accelerate library. The model is trained using the AdamW optimizer with a linear learning rate schedule, a learning rate of $5 \times 10^{-5}$, and 500 steps of warmup. Following recommendations for training on large datasets (Muennighoff et al., 2025), we train for a total of 4 epochs.

Due to compute constraints, different model sizes in memory, and different dataset sizes implied by Chinchilla scaling laws, our training setups vary slightly between model sizes. We summarize these variations in Table 4.

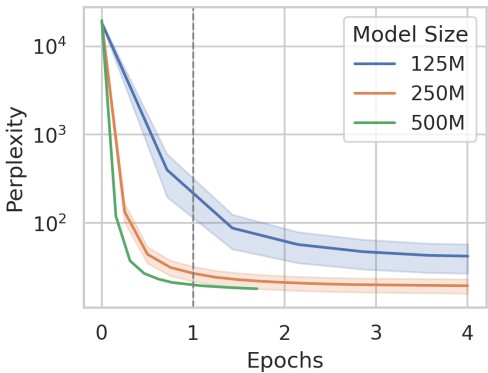

Figure 11: Out-of-sample Perplexity vs Epochs for all TradeFM models. Epoch 1 represents one pass over the entire dataset; subsequent training is on repeated data.

## B.4 TOKENIZER PSEUDOCODE

**Algorithm 1** Tokenizer Pseudocode

---

**Input:** Flattened trading data with features: price, volume, time, action, side, trader, account
**Input:** Bin counts: $N_{\text{price}}$, $N_{\text{depth}}$, $N_{\text{volume}}$, $N_{\text{time}}$, $N_{\text{side}} = 2$, $N_{\text{action}} = 2$
1: **for** each feature $f$ in {price, depth, volume, time} **do**
2:    Remove NaN and infinite values from $f$
3:    Compute upper outlier threshold $u = \text{percentile}(f, 99)$
4:    **if** feature is double-sided **then**
5:        Compute lower outlier threshold $l = \text{percentile}(f, 1)$
6:        Assign values outside $[l, u]$ to lower / upper outlier bins
7:    **else**
8:        Assign values above $u$ to upper outlier bin
9:    **end if**
10:    **if** using equal-frequency bins **then**
11:        Compute bin edges $B_f$ using quantile binning with $N_f$ bins
12:    **else**
13:        Compute bin edges $B_f$ using histogram binning with $N_f$ bins
14:    **end if**
15:    Digitize $f$ into bin indices $I_f$ using $B_f$
16:    Handle outliers: assign out-of-range values to edge bins as needed
17: **end for**
18: **for** each categorical feature $c$ in {action, side, trader, account} **do**
19:    Map each category to a unique integer index $I_c$
20: **end for**
21: **for** each order $o$ in the dataset **do**
22:    **for** each feature $f$ **do**
23:        **if** $o[f]$ is NaN **then**
24:            Impute $o[f]$ with a random valid bin index or default value
25:        **end if**
26:    **end for**
27:    Compute token index for $o$:
28:    $T_o = I_{\text{action}} \times N_{\text{side}} \times N_{\text{depth}} \times N_{\text{volume}} \times N_{\text{time}}$
29:        $+ I_{\text{side}} \times N_{\text{depth}} \times N_{\text{volume}} \times N_{\text{time}}$
30:        $+ I_{\text{depth}} \times N_{\text{volume}} \times N_{\text{time}}$
31:        $+ I_{\text{volume}} \times N_{\text{time}}$
32:        $+ I_{\text{time}}$
33:    Assign $T_o$ to order $o$
34: **end for**

---

972
973
974
975
976
977
978
979
980
981
982
983
984
985
986
987
988
989
990
991
992
993
994
995

## B.5 MARKET SIMULATION PSEUDOCODE

996
997
998
999
1000
1001
1002
1003
1004
1005
1006
1007
1008
1009
1010
1011
1012
1013
1014
1015
1016
1017
1018
1019
1020
1021
1022
1023
1024
1025

---

**Algorithm 2** Market Simulator: Part 1 - Initialization and Utilities

---

**Input:** Sequence of order transactions, initial price $p_0$, simulation parameters
1: **Initialize Exchange:**
2: Set initial price $p_0$
3: Initialize order book, midprice, fills, deletes, spreads, bid/ask volumes

4: **Function:** INITIALIZEORDERBOOK(order_columns)
5: Reset order book, midprice, fills, deletes, spreads, bid/ask volumes
6: Set initial bid/ask to $p_0$

7: **Function:** GETORDERPRICE(transaction)
8: **if** order is market **then**
9:    **if** side is Sell **then**
10:      price ← lowest ask
11:    **else**
12:      price ← highest bid
13:    **end if**
14: **else**
15:    price ← (order price depth / 10,000) × current midprice + current midprice
16: **end if**
17: **Return** price

18: **Function:** GENERATEFILL(best_past_order, order, quantity)
19: Compute time since best_past_order
20: Determine match price:
21: **if** both orders are market **then**
22:    price ← current midprice
23: **else if** order is market **then**
24:    price ← best_past_order price
25: **else if** best_past_order is market **then**
26:    price ← order price
27: **else**
28:    price ← best_past_order price
29: **end if**
30: **Return** fill record with IDs, sides, prices, depths, volume, time delta

---

---

**Algorithm 3** Market Simulator: Part 2 - Simulation Step Functions

---

31: **Function:** STEPORDERBOOK(order)
32: Extract side and price from order
33: **while** order volume $> 0$ **do**
34:     **if** side is Sell **then**
35:         Find matching Buy orders with price $\geq$ order price
36:     **else**
37:         Find matching Sell orders with price $\leq$ order price
38:     **end if**
39:     **if** no matching orders **then**
40:         **break**
41:     **end if**
42:     Select best matching order (highest bid or lowest ask)
43:     **if** best_past_order volume $>$ order volume **then**
44:         Reduce best_past_order volume by order volume
45:         Record fill and **return**
46:     **else if** best_past_order volume $<$ order volume **then**
47:         Reduce order volume by best_past_order volume
48:         Remove best_past_order from book
49:         Record fill
50:     **else**
51:         Record fill
52:         Remove best_past_order from book
53:         **return**
54:     **end if**
55: **end while**
56: **if** order volume $> 0$ **then**
57:     Add partially filled order to book
58: **end if**

59: **Function:** STEPMIDPRICE(transaction)
60: **if** transaction is Delete **then**
61:     Use current order book
62: **else**
63:     Add transaction to temporary order book
64: **end if**
65: Update highest bid and lowest ask from book
66: Compute midprice as average of highest bid and lowest ask
67: Record midprice and bid/ask volumes

68: **Function:** STEPSIM(transaction)
69: Update transaction midprice
70: **if** action is Add **then**
71:     Compute order price
72:     Update midprice
73:     Step order book
74: **else if** action is Delete **then**
75:     Match on order ID
76:     Remove matching orders and record deletes
77:     Update midprice
78: **end if**
79: Record simulation time for profiling

80: **Function:** RUNSIMULATION(data)
81: Initialize order book
82: **for** each transaction in data **do**
83:     StepSim(transaction)
84: **end for**
85: **Return** fills and midprice history

---

### B.6 ZERO-INTELLIGENCE BASELINE

The Zero-Intelligence (ZI) agent is a canonical null model used to test whether a model learns complex, conditional dynamics beyond the market's basic structural properties (Gode & Sunder, 1993; Farmer et al., 2005). To provide a fair baseline, our ZI agent generates orders stochastically by sampling from distributions calibrated to match the marginals of key features in a 450-million-trade sample of the training data.

Specifically, side and action type are sampled from their empirical categorical distributions; interarrival time is sampled from a fitted Exponential distribution; order volume from a fitted Exponential distribution; and price depth is drawn from a Gaussian Mixture Model (GMM).

The resulting ZI agent orders are processed through the identical market simulator and evaluation pipeline as TradeFM to ensure a direct and fair comparison. We compute 2,048 rollouts of 1,024 events, and compute the same stylized facts.

### B.7 COMPOUND HAWKES BASELINE

Hawkes Processes are commonly applied to market data for their ability to robustly model interarrival times of self-exciting events (Bacry et al., 2015; Jain et al., 2024). We adopt the Compound Hawkes model which combines a Hawkes process for modeling interarrival times with empirical distributions for modeling additional event features like volume and price depth. We use the same 450-million-trade data as is used to train our zero-intelligence baseline, and separate the data based on action and side.

We then fit a Hawkes process using a sum of exponential kernel, with 4 dimensions, one for each combination of action and side (buy-delete, buy-add, sell-delete, sell-add). For each of these action-side combinations we calibrate a Gaussian Mixture Model for price depths, and an Exponential for volume.

## C SCALING ANALYSIS

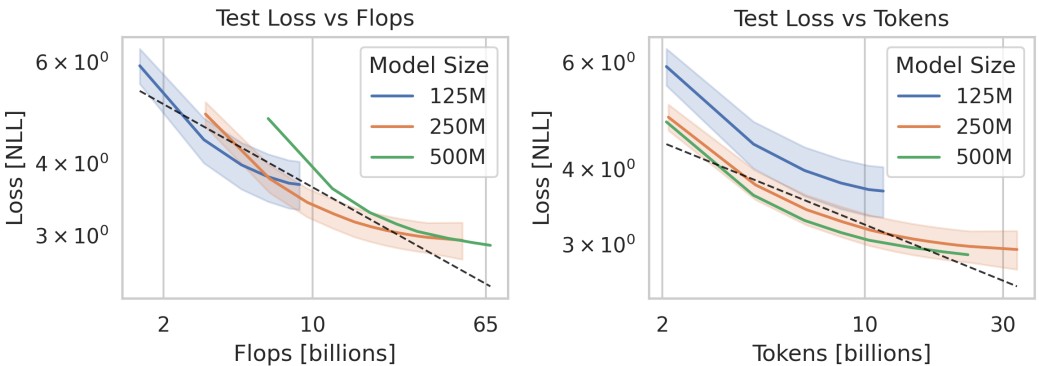

Figure 12: Scaling law results reporting test loss (negative log likelihood) on held-out data one month in advance of the training data cutoff. The black dashed line represents the power law fit to the minimum loss frontier.

To substantiate the *Foundation Model* claim, we conducted a comprehensive scaling analysis of our approach. We trained models $\in \approx [125M, 250M, 500M]$.

Our **500M parameter model** is still training, and while currently at an early checkpoint, its performance aligns with our scaling projections. The scaling law plots in Figure 12 demonstrate the expected power-law relationships between compute, dataset size, and test loss. These plots include repeated data, as we train for four epochs. We verify this by computing the minimum loss frontier in terms of both compute and dataset size, and fitting power laws to find that the test loss $L(C)$ with respect to compute in Flops $C$, and $L(D)$ with respect to dataset size in tokens $D$, follow:

$$L(C) \propto C^{-\alpha_C}; \alpha_C \approx 0.21$$
$$L(D) \propto D^{-\alpha_D}; \alpha_D \approx 0.19$$

While 500M is small relative to general purpose LLMs (Llama-3 8B, GPT-OSS 20B), it is large for the Financial Microstructure domain, and similar to other domain-specific models such as MaRS. Standard SOTA models in this field typically have $10M$ parameters (e.g., DeepLOB (60K params) and LOBS5 (6.3 M params)). TradeFM represents a $> 50x$ **increase in model capacity** over existing domain-specific baselines.

# D EXTENDED EXPERIMENTAL RESULTS

## D.1 DATASET DETAILS

Table 5 contains details on the various held-out datasets used for evaluation.

| Country | Number of Assets | Date-Asset Pairs | Tokens |
|---|---|---|---|
| US | 6,885 | 81,203 | 857,017,219 |
| China | 4,926 | 68,925 | 37,408,529 |
| Japan | 2,932 | 37,235 | 286,476,052 |

Table 5: Dataset statistics for US, China, and Japan held-out data. All geographies are evaluated on Jan. 2025 data.

## D.2 SIMULATOR VALIDATION

In order to evaluate the simulator, we replay sequences of real orders through it and compare the statistical properties of the resulting simulated trade fills against the real fills from our historical data. We focus on two key metrics: the cumulative distribution function (CDF) of fill volumes and the CDF of lot counts (the number of discrete fills required to complete a single order). As shown in Figure 13, we find a strong correspondence between the real and simulated distributions across assets of varying liquidity, confirming that our simulator is a high-fidelity environment for evaluation. We find correlations of 0.91 and 0.98 between sim and real volumes and lot counts, respectively.

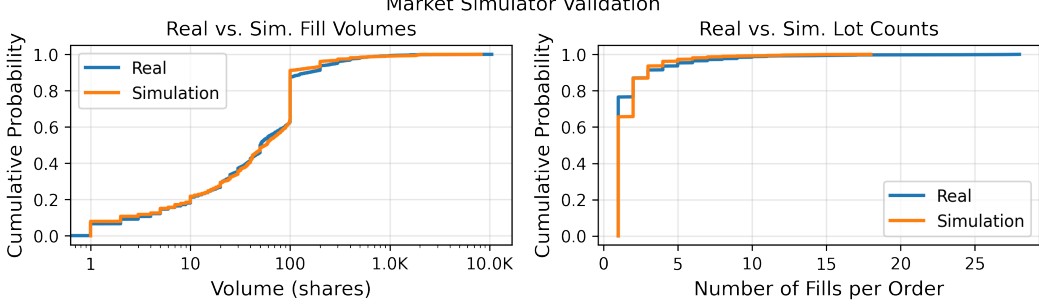

Figure 13: Stylized facts of market simulator fills: (left) fill volumes; (right) lot counts, or number of separate fills taken to fulfill an order. We see good correspondence between simulated and actual fills, with a correlation of 0.91 for volumes and 0.98 for lot counts, respectively.

## D.3 TEMPORAL DRIFT

As financial markets are dynamic and market regimes are constantly changing, we investigate the tendency of model performance to drift over time. Our tokenizer's main contribution is to standardize representations of market features over both the liquidity and time regime.

In Fig. 14 we demonstrate the universality of these features by exploring the distribution of our relative price level, relative price depth, interarrival time, and volume features in both the month used to calibrate our tokenizer, Feb. 2024, and one year later in Feb. 2025. We observe that our features are stationary over this period even as volatility, price level, and other market conditions vary. Fig. 15 shows the Kolmogorov-Smirnov and Wasserstein distance of each of these features between the tokenizer calibration month and each of 9 held-out months. We include a non-stationary feature, raw midprice, to contextualize the stationarity of these metrics.

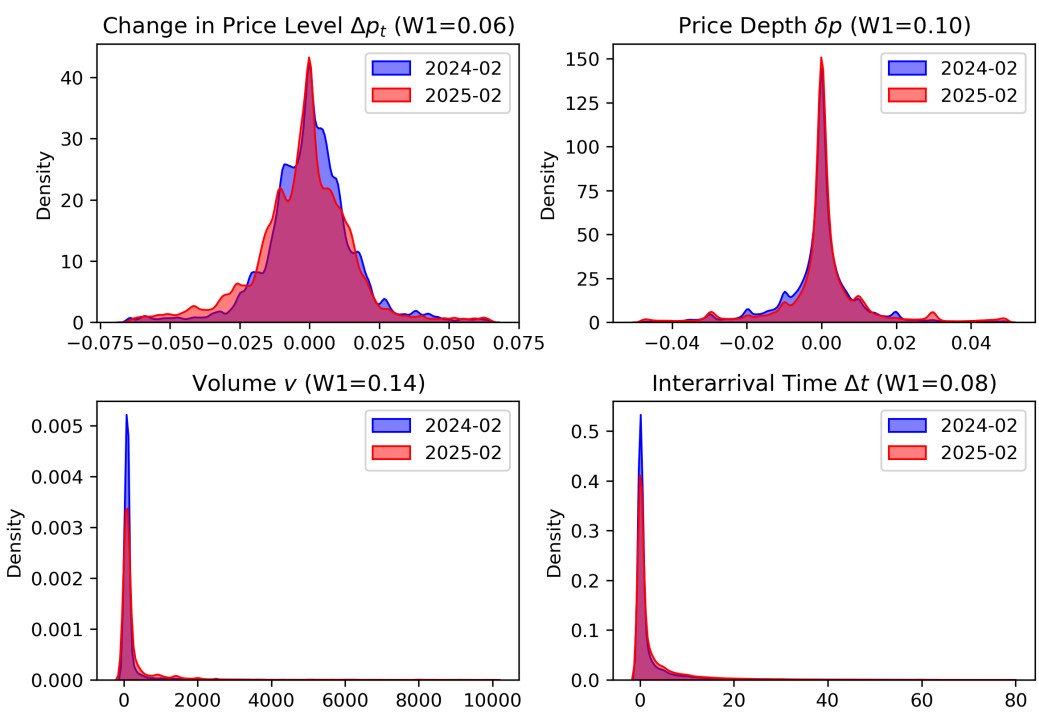

Figure 14: Kernel-density estimation of feature distributions from the tokenizer calibration period of Feb. 2024 to one year later in Feb. 2025. Our feature engineering successfully makes these features stationary over time, allowing our model to generalize to out of distribution temporal regimes.

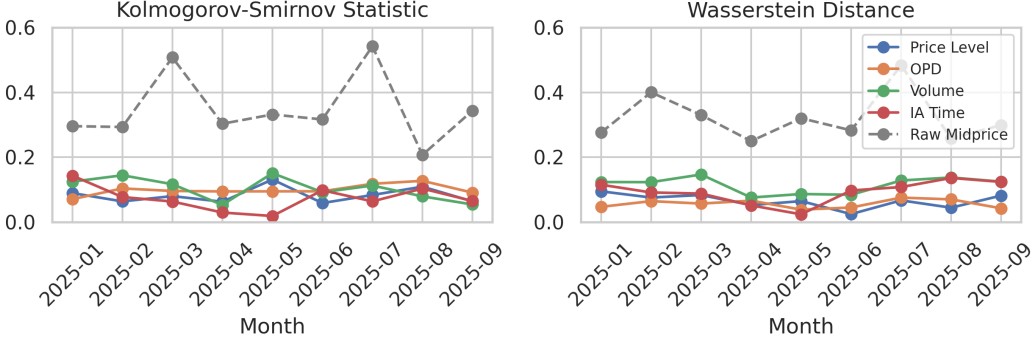

Figure 15: Kolomogorov-Smirnov and Wasserstein distances between distributions of our features during our tokenizer calibration month and held-out months. We include raw midprice, a non-stationary feature, for context.

In Fig. 16 we extend the aggregated results in Table 2 for all quantities of interest. We observe that while these metrics do vary within a range, the variance is small and our method mostly achieves higher fidelity than baselines.

### D.4    MARKET SIMULATION & STRESS TESTING

The integrated TradeFM-simulator system functions as a high-fidelity environment for complex "what-if" analyses and stress testing. This allows for the study of systemic risk and market sta-

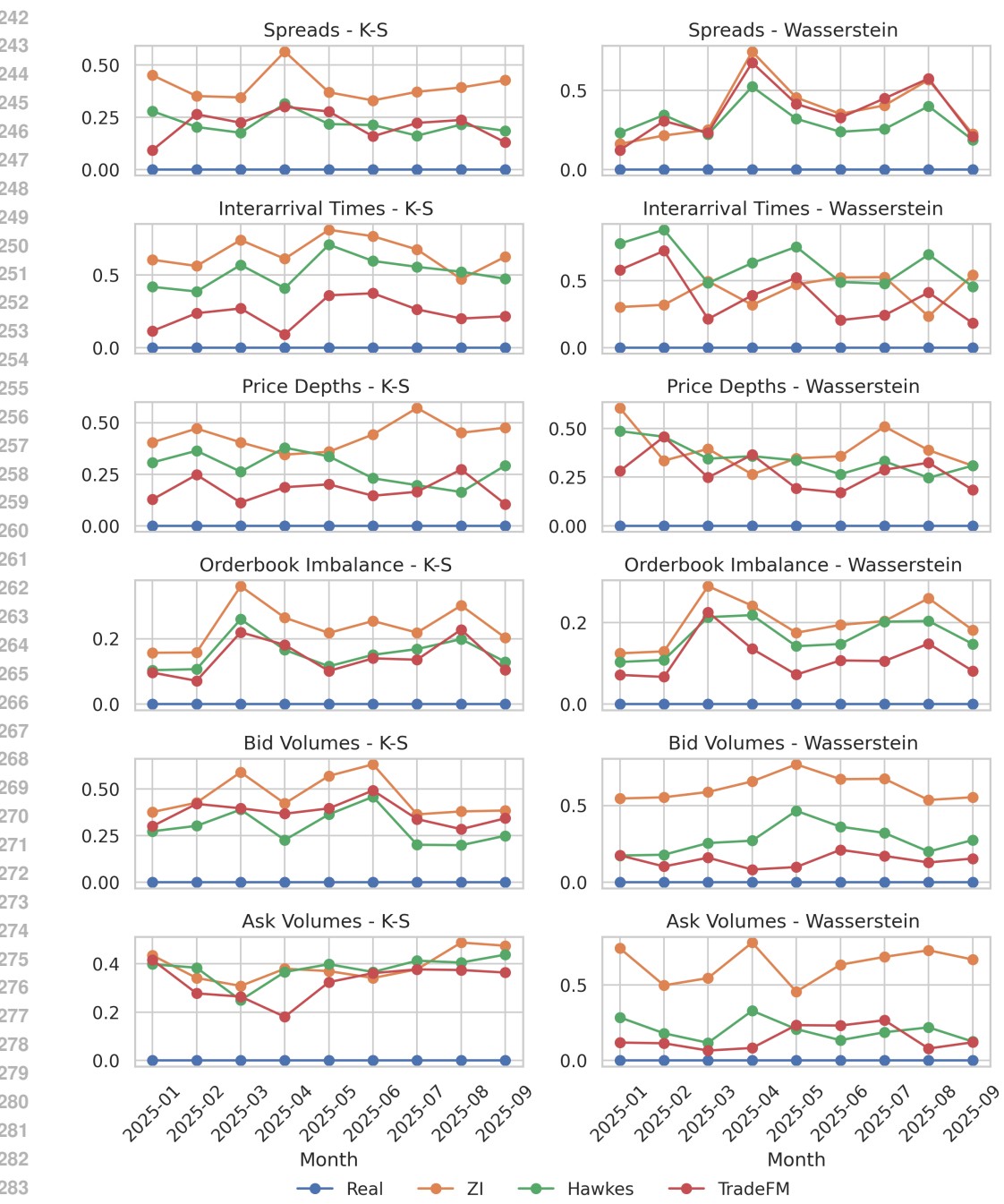

Figure 16: Wasserstein distance and Kolmogorov-Smirnov statistic of feature distributions and emergent market factors from various methods over nine held-out months.

bility in a controlled environment. The ability to generate plausible, multi-step forecasts of future market trajectories, as illustrated in Figure 17, is a direct outcome of this closed-loop simulation capability.

Such systems are also useful to regulators and risk managers (Dwarakanath et al., 2024), who can use this system to simulate the market's response to extreme or counterfactual scenarios, such as by injecting large, anomalous orders into a historical context and observing the resulting price tra-

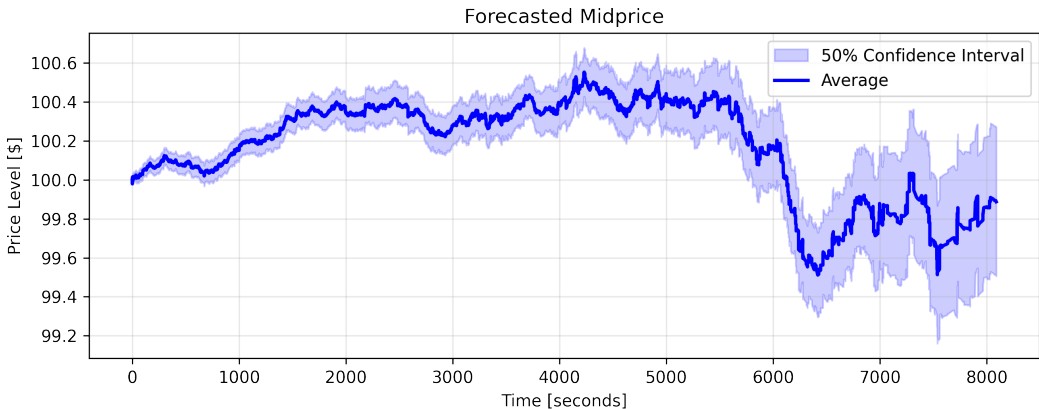

Figure 17: Multi-step mid-price forecast generated via rollouts for an imaginary asset initially priced at $100. The average trajectory and 50% confidence interval over 256 simulations show the model generates plausible, non-stationary market paths.

jectory. Fig. 18 demonstrates this capability – for a sample asset, we artificially inject buy or sell orders at 10x the frequency found in the real context, and average midprice forecasts over 10 rollouts. When we inject artificial sell orders, the midprice drops, and when we inject buy orders, the midprice rises, illustrating realistic behavior.

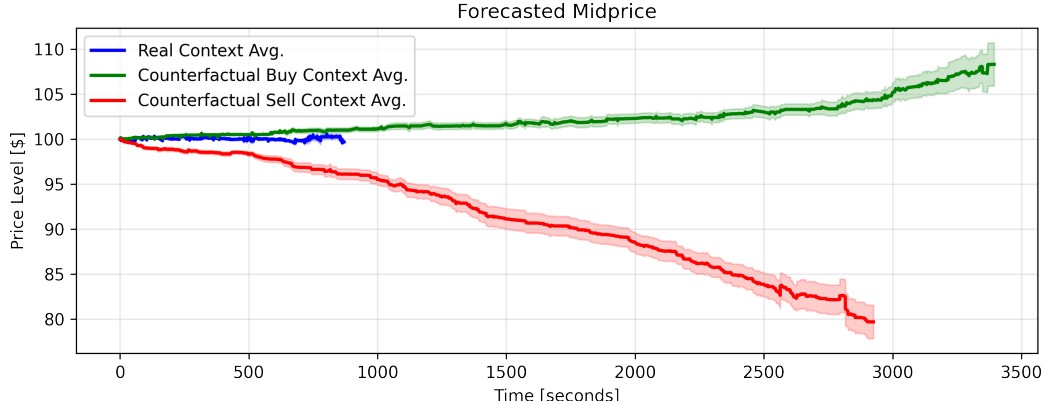

Figure 18: Stress testing via counterfactual simulation. The model's generated price path responds realistically to injected anomalous order flow (10x normal frequency), demonstrating its utility for impact analysis.

### D.5 MULTI-AGENT MODELING & RL FINE-TUNING

Our system provides a high-fidelity, interactive environment for training and evaluating sophisticated, learning-based agents. The pre-trained TradeFM can serve as a realistic "background" market, generating plausible and reactive counterparty order flow. This creates a dynamic training ground for:

- **Reinforcement Learning (RL) for Optimal Execution**: RL agents can be trained to learn optimal strategies for executing large orders by interacting with the simulated market, minimizing costs such as price impact and the bid-ask spread.
- **Multi-Agent Systems (MAS)**: The simulator can be populated with multiple, heterogeneous learning-based agents to study the emergent, collective behaviors and potential instabilities that arise from their interactions. The participant-level conditioning of our model provides a natural and powerful mechanism for initializing and fine-tuning diverse agent policies within such a system.

