# OpenReview forum: "TradeFM: A Generative Foundation Model for Trade-flow and Market Microstructure"
_ICLR.cc/2026/Conference — Submitted to ICLR 2026_

### Official Review · Reviewer_aZyA · 2025-10-29

**Soundness:** 2
**Presentation:** 3
**Contribution:** 2
**Rating:** 4
**Confidence:** 4

**Summary:**

The paper introduces TradeFM, a large-scale generative foundation model designed for cross-asset financial market modeling. It addresses the challenge that traditional models are often asset-specific and fail to generalize across different market microstructures.

**Strengths:**

This paper is generally well written and easy to follow. However, several design choices appear somewhat ad hoc and lack clear theoretical justification or motivation. Without a stronger conceptual grounding, it is difficult to judge whether these modeling decisions are fundamentally sound or simply work well empirically.

**Weaknesses:**

1. The dataset used in the study is proprietary and not publicly released, which hinders independent replication and validation of the results.


2. The paper only compares TradeFM with a zero-intelligence random trader baseline. Although classical models such as Hawkes processes and early deep learning order-book models (e.g., DeepLOB) are mentioned, they are not empirically evaluated, weakening the comparative validity of the claims.

3. The paper lacks systematic quantitative evaluation of out-of-distribution generalization — e.g., performance on unseen assets, extreme volatility regimes, or temporal shifts — relying mostly on qualitative or intuitive arguments.


4. The market replay experiments simulate relatively short event sequences (512 trades per rollout). The paper does not examine whether the model can stably reproduce long-term market dynamics across multiple trading days.


5. While the empirical design of TradeFM is sound, the paper does not provide sufficient theoretical grounding for why the chosen feature normalization, tokenization, and model architecture should guarantee cross-asset generalization or preserve market invariants. A more formal discussion linking design choices to market microstructure theory would strengthen the work.

**Questions:**

1. Could the authors include comparisons with traditional models such as Hawkes process–based methods or deep learning order book models like DeepLOB, to better contextualize the empirical performance of TradeFM?

2. Could the evaluation of generalization be made more rigorous, for example by conducting cross-asset tests, rolling-window experiments, or stress simulations under different market regimes?

3. Is there any plan to release the tokenizer design or dataset, at least partially, to improve reproducibility and facilitate further research on cross-asset generative modeling?

---

> ### Author Response · Authors · 2025-12-02
> **Response to Reviewer 4 (aZyA)**
>
> We thank the reviewer for their constructive feedback and for finding the paper well written and easy to follow. We appreciate your push for more rigorous baselines and generalization tests, which has significantly strengthened the paper.
>
> ### W1: "The dataset is proprietary... hinders independent replication."
>
> We acknowledge this constraint. To mitigate it, we have **added Appendix B (Reproducibility Guide)** which includes detailed pseudocode and hyperparameter tables. This allows the community to replicate our methodology on open datasets.
>
> ### W2/Q1: "Only compares TradeFM with a zero-intelligence random trader baseline... Could the authors include comparisons with traditional models such as Hawkes process–based methods... to better contextualize the empirical performance."
>
> This is a great suggestion. We have **added a Compound Hawkes Process baseline** to benchmark performance against standard method. As shown in **Tables 1 and 2**, our method demonstrates closer fidelity to real data in terms of log return distributions across return intervals and distributions of various emergent facts like spreads, orderbook imbalances, bid and ask volumes, and more.
>
> We selected the Hawkes method as our primary baseline because it is the canonical generative model for order flow dynamics. While discriminative deep learning methods (such as DeepLOB) are powerful models for price prediction, they do not naturally generate continuous-time multi-feature order flow sequences, making Hawkes the more direct and rigorous comparison for our specific generative task.
>
> ### W3/Q2: "Lacks systematic quantitative evaluation of out-of-distribution generalization..."
>
> Thank you for the excellent suggestions. We have **significantly expanded our OOD evaluation** in the revision:
>
> 1. Temporal OOD: We evaluate on Jan - Sep 2025 data (a period of higher volatility than the training set). Our 500M parameter model maintained an average perplexity across these months of 23.41, compared to a slightly lower in-sample perplexity of 20.91, demonstrating robustness to regime shifts.
>
> 2. Geographic OOD: We also measured the model's zero-shot generalization capabilities by measuring performance on completely held-out assets. Figure 5 shows US performance compares to Japan and China markets, demonstrating that the model's universal tokenization generalizes to completely unseen market structures.
>
> ### W4: "Market replay experiments...short event sequences (512 trades per rollout)... does not examine... long-term market dynamics."
>
> We have **doubled our standard evaluation rollout length** to 1,024 steps to further validate stability over longer horizons. While this length was selected to balance compute requirements with rigorous evaluation, it **does not represent the maximum capability of the model**. As demonstrated in Figure 17, the model maintains stability for multi-hour rollouts.
>
> We emphasize that high-frequency order-flow models are fundamentally designed for **intraday dynamics**; simulating multi-day horizons is generally outside the scope of microstructure models due to the structural reset of the limit order book at the end of each trading session.
>
> ### W5: "While the empirical design of TradeFM is sound...discussion linking design choices to market microstructure theory would strengthen the work."
>
> We have expanded Section 5.3 to explicitly link our scale-invariant features to the **theory of "Universal Price Formation"** (Sirignano \& Cont, 2021). This theory posits that while asset prices vary widely, the underlying mechanics of price formation -- driven by the interplay of order flow, liquidity, and volatility -- are universal and invariant across assets when properly normalized.
>
> Our subsequent design choices in normalization and tokenization enable TradeFM to learn the invariant grammar of market microstructure, providing a strong theoretical basis for its cross-asset generalization capabilities.
>
> ### Q3: "Any plan to release the tokenizer design or dataset, at least partially..."
>
> While strict industrial IP constraints prevent the release of proprietary data, we are fully committed to enabling reproducibility. **We have released the tokenizer design and calibration logic** in the new Appendix B (Reproducibility Guide). This ensures that the core methodological contribution -- the cross-asset generative modeling framework -- is open and reproducible for the community.

---

### Official Review · Reviewer_nB6E · 2025-10-30

**Soundness:** 2
**Presentation:** 2
**Contribution:** 2
**Rating:** 4
**Confidence:** 4

**Summary:**

This paper proposes TradeFM, a decoder-only Transformer for modeling market microstructure as an autoregressive sequence over tokenized trade events. The major contributions include: (i) a novel tokenization that compresses multi-feature trade tuples into a single token, (ii) closed-loop rollouts with a market simulator, and (iii) high-fidelity synthetic data.

**Strengths:**

- Universal representation learned by TradeFM seems useful in downstream applications.

- The proposed approach directly works on raw data, without the need for human expertise.

**Weaknesses:**

- The introduction of scale-invariant features in Section 5.3  is unclear for readers who are not familiar with this concept. Concrete, self-contained explanations and citations to prior works are needed. Personally, I prefer to involve some simple illustrations or examples to clarify this.

- This paper only evaluates the fidelity of synthetic data generation using stylized facts. To evaluate the fidelity of synthetic time series, there are many other metrics, including goodness of fit and experimental settings like “train on synthetic, test on real“ (TSTR) [1].

- No ablation study to verify the effectiveness of the proposed tokenization and scale-invariant features.

[1] Yoon et al. "Time-series generative adversarial networks." *Neurips* 2019.

**Questions:**

- How can we define and interpret the universal representation in this context? Can we use this as a pre-trained embedding in other deep learning models?

- What exactly is the pre-training task? Next-token prediction?

- Is there a risk of data leakage using the train-test split?

- What’s the impact of different setups of bin type and size?

- There is only one baseline compared in the experiments. Are there any more recent ones?

- What’s the data source? Is it public?

---

> ### Author Response · Authors · 2025-12-02
> **Response to Reviewer 3 (nB6E)**
>
> We thank the reviewer for the constructive assessment and for recognizing that our approach "directly works on raw data" without human feature engineering. We also appreciate the reviewer acknowledging the potential impact of our work in downstream applications.
>
> ### W1: "The introduction of scale-invariant features... unclear for readers who are not familiar with this concept.  Concrete, self-contained explanations...needed."
>
> We apologize for the lack of clarity. We have revised Section 5.3 and include a step-by-step walkthrough for transforming a raw trade tuple into our scale-invariant coordinates in Appendix A.4.
>
> ### W2: "Evaluates... only using stylized facts... other metrics, including goodness of fit."
>
> This is an excellent suggestion. We have added **Tables 1 and 2** which report the Wasserstein Distance and Kolmogorov-Smirnov (K-S) statistics for generated distributions of Log Returns (varying return intervals), Volume, Spread, and Interarrival Times. These metrics provide the rigorous quantitative goodness-of-fit assessment requested. Our method produces log return and emergent property distributions that exhibit higher fidelity to real data than all other approaches.
>
> ### W3: "No ablation study to verify... tokenization and scale-invariant features."
>
> We provide a theoretical ablation and empirical evidence. Raw price inputs vary by orders of magnitude ($10^0$ to $10^5$). Without scale invariance, a single shared vocabulary cannot represent both penny stocks and high-priced assets. **Figure 8** demonstrates how relative price features normalize distributions across liquidity tiers. **Figures 14 and 15** show feature stability across time.
>
> ### Q1: "How can we define and interpret the universal representation..."
>
> We define the "universal representation" as the learned latent space which captures market dynamics independent of asset-specific traits such as nominal price or liquidity tier.
> Yes, these learned representations can be frozen and used as rich context-aware embeddings for downstream tasks such as forecasting or trade classification. We have updated Section 10 (Applications) to emphasize this capability.
>
> ### Q2: "What exactly is the pre-training task? Next-token prediction?"
>
> Yes. As stated in Section 4, the pre-training task is autoregressive next-token prediction (akin to Causal Language Modeling). The target token is the composite trade event token ($i_{trade}$) derived from our mixed-radix tokenization scheme.
>
> ### Q3: "Is there a risk of data leakage using the train-test split?"
>
> **No.** We employ a strict **temporal hold-out strategy** to prevent look-ahead bias, ensuring that the model never sees future data during training, mimicking a real-world deployment scenario. For example, for our 500M parameter model, our tokenizer is calibrated on the first 30 days of data in Feb. 2024, our training data covers Feb. 2024 through Dec. 2024, and our test data covers Jan. through Sept. 2025.
>
> ### Q4: "What’s the impact of different setups of bin type and size?"
>
> Our **hybrid binning strategy** is designed for specific statistical properties of market features. We use **Equal-Frequency binning for price features** to maximize resolution in the dense center of the distribution. We use **Equal-Width binning on log-transformed volume and interarrival time** to capture the heavy-tailed, power-law dynamics; Uniform binning fails to capture these tails effectively.
>
> ### Q5: "There is only one baseline compared in the experiments. Are there any more recent ones?"
>
> We acknowledge this limitation in the initial submission. In this revision, we have added a **Compound Hawkes** baseline – the standard generative model for high-frequency order flow in econometrics. In contrast, models like DeepLOB are primarily discriminative (focused on price prediction) rather than generative, and other generative formulations (e.g., MaRS, LOBS5) often require full limit-order-book snapshots which differs from our partial observability setting.
>
> ### Q6: "What’s the data source? Is it public?"
>
> The model is pre-trained on a large-scale proprietary dataset of US equities transactions (Section 5.1). While we cannot release this specific industrial dataset, we have added **Appendix B (Reproducibility Guide)** which contains the complete methodology, including the tokenizer and simulator logic. This ensures that the method is reproducible on public datasets such as LOBSTER.

---

### Official Review · Reviewer_E5tj · 2025-10-31

**Soundness:** 2
**Presentation:** 2
**Contribution:** 2
**Rating:** 2
**Confidence:** 4

**Summary:**

The paper introduces TradeFM to model financial market microstructure under partial observability. The authors consider scale-invariant features and propose a new tokenization scheme. The model is validated via a closed-loop simulator.

**Strengths:**

- The research problem is critical and challenging in financial economics.

- Design choices reflect market pragmatism. The model is learned from partial observations to mimic the realistic scenarios, which aligns with how practitioners actually see the market.

**Weaknesses:**

- Tokenizer calibration on the first 30 days invites regime bias and drift.

- Validation for the quality of synthetic data using stylized facts is not convincing enough. For example, the lack of autocorrelation and heavy tails are easy for many generative models to achieve, but their practical usefulness is limited in financial applications.

- Baselines are limited. A zero-intelligence agent is a weak competitor.

- There is no strong evidence that synthetic data improves downstream tasks.

**Questions:**

- This work integrates the idea of scale-invariant features in financial data with autoregressive modeling. Since scale invariance implies recurring patterns across time, have you considered modeling market data using a pattern-based scheme?

- What do you observe about how LLMs understand tokenized transaction data?

- Why is the tokenizer calibrated on the specific first 30 days of data?

---

> ### Author Response · Authors · 2025-12-02
> **Response to Reviewer 2 (E5tj)**
>
> We appreciate the reviewer's recognition that our design choices reflect market pragmatism and align with practitioner perspectives. We have leveraged your feedback to significantly strengthen our evaluation rigor.
>
> ### W1: "Tokenizer calibration on the first 30 days invites regime bias and drift."
>
> We empirically validate this in **Figure 14 and Figure 15 (Appendix D)**. We compute Wasserstein distances between feature distributions during calibration (Feb 2024) and each of 9 held-out months beginning one year later (Feb 2025). We see minimal distribution shift, confirming our scale-invariant features effectively "stationarize" the data. We show the real distributions of our features for one month in particular in Figure 14.
>
> ### W2: "Validation... using stylized facts is not convincing enough... lack of autocorrelation and heavy tails are easy \[to achieve\]."
>
> We agree that relying solely on stylized facts can be insufficient. To address this:
> - We compute **log-return distributions** across various return intervals ($\Delta t_r$) for all methods and compare to real data. We report the Wasserstein Distance and Kolmogorv-Smirnov (K-S) statistics in **Table 1** and find that our method consistently produces log return distributions closer to real data, across a range of return intervals.
> - We have conducted a **rigorous quantitative evaluation of the generated order flow distributions**, inspired by the recent LOB-Bench framework (Nagy et al., 2025). We report the Wasserstein and K-S distances for key order flow properties like spreads, orderbook imbalance, order price depths, and more in **Table 2**, over nine months of held-out data from Jan - Sep 2025. Our method demonstrates consistent distributional fidelity closer to real than baselines, even during high-volatility periods.
>
> ### W3: "Baselines are limited. A zero-intelligence agent is a weak competitor."
>
> We concede that the previous Zero-Intelligence was insufficient. We have added a Compound Hawkes baseline (Bacry et al., 2015, Jain et al., 2024). As shown in **Tables 1 and 2**, TradeFM captures log return distributions and emergent distributions of order flow like spreads and orderbook imbalance significantly better than Hawkes.
>
> ### W4: "There is no strong evidence that synthetic data improves downstream tasks."
>
> As the primary focus of this paper is Generative Fidelity, we provide evidence of utility via **Distributional Fidelity** of second-order generative rollouts (model + simulator). Proving downstream alpha generation is outside the scope of this study, though we lay the groundwork for it in **Section 10**.
>
> ### Q1: "Have you considered modeling market data using a pattern-based scheme?"
>
> We view the Transformer architecture itself as a general-purpose "pattern-based scheme". The self-attention mechanism allows the model to dynamically attend to sequential patterns in order flow history to predict the next event. We have also looked at this problem from the lens of more traditional ML based pattern matching techniques, we will add those citations in the final version (currently omitted to preserve anonymity).
>
> ### Q2: "What do you observe about how LLMs understand tokenized transaction data?"
>
> We observe the model learns market concepts that generalize across geographies. As shown in **Figure 5**, the model performs well on Asian markets (Japan, China) despite being trained primarily on US data. This suggests that it *understands* universal microstructure mechanics rather than simply memorizing US-specific asset behaviors.
>
> ### Q3: "Why is the tokenizer calibrated on the specific first 30 days of data?"
>
> Calibrating on the initial period is a standard practice to prevent data leakage; it simulates the realistic scenario wherein a model must operate on future data using a vocabulary defined by past data. As noted in our response to W1 and demonstrated in **Figures 14 and 15**, this does not lead to drift, as the scale-invariant features ensure the token distribution remains stable over time.

---

### Official Review · Reviewer_JSa7 · 2025-11-01

**Soundness:** 1
**Presentation:** 2
**Contribution:** 2
**Rating:** 2
**Confidence:** 4

**Summary:**

The paper introduces TradeFM, a generative foundation model for trade-flow and market microstructure, pre-trained on billions of equities transactions over 130 trading days across 8,365 assets to learn a unified representation without asset-specific calibration. It learns from a partially observed market state, uses a scale-invariant feature representation with Exponentially-Weighted Volume-Weighted Average Price (EW-VWAP) mid-price estimation, and applies a universal tokenisation scheme that maps multi-feature events into a composite integer via a mixed radix construction (vocabulary 16,384), conditioned on liquidity, ∆p, and IMP. The architecture is a decoder-only Transformer based on the Llama family with grouped-query-attention (GQA) and rotary positional encoding (RoPE), totalling 251M parameters and achieving a held-out perplexity of 17.85. Evaluation couples TradeFM with a deterministic market simulator in a closed-loop rollout using price-time priority matching, where generated events update the LOB state that is then fed back to the model. Simulations reproduce stylised facts—lack of autocorrelation in returns, volatility clustering (slowly decaying ACF of absolute returns), heavy tails, and aggregational Gaussianity—and outperform a calibrated Zero-Intelligence (ZI) baseline on metrics such as kurtosis. The conditioning signals (liquidity tiers and market-level vs. participant-level indicators) enable controllable generation, supporting high-fidelity synthetic data generation, backtesting, data augmentation for illiquid assets, and multi-agent modelling.

**Strengths:**

The choice of a good research topic, Foundation Model, is a significant advantage for the paper. A clear formulation of trade-flow modelling under partial observability, including an EW-VWAP mid-price estimator and a scale-invariant feature representation that aims to align assets across liquidity regimes. A universal tokenisation via mixed-radix composite trade tokens (vocab 16,384) plus conditioning signals (liquidity bins, Δp, IMP), which yields a compact tabular embedding pipeline. A closed-loop evaluation with a deterministic price–time-priority simulator; the generated rollouts reproduce key stylised facts (near-zero return ACF, volatility clustering, heavy tails, aggregational Gaussianity). Controllability via conditioning (market- vs participant-level; liquidity tiers) showing intuitive shifts in volume/interarrival variability.

**Weaknesses:**

(1) Scaling evidence is missing for a foundation model claim. The paper reports a single model size (251M) and a single held-out perplexity (17.85). Please provide reproducible scaling curves across model sizes, data fractions, and compute (e.g., 50M/150M/251M/500M; 25/50/100% tokens), with monotonic trends and power-law fits. Without multi-point fits, the FM positioning is under-supported.

(2) Novelty vs MaRS and strong baselines. The paper positions universality largely at the feature/tokenization level; however, prior work (e.g., MaRS) also trains across heterogeneous assets.

(3) Spanning 130 trading days from July 2024 to January 2025, across 8,365 unique assets, the data scale is not large, especially compared to MarS and LOBS5 (LOB Bench), because U.S. equity trading volume is concentrated in the front, high-liquidity names. For example, in the LOBSTER data, Apple’s one-year message and order book data, after compression, is 64 GB; even for S&P 500 stocks with low liquidity, the compressed data size is only about 20 MB.

(4) Many passages appear to be AI-generated. However, I did not see a disclaimer in the paper indicating this. I may be mistaken, but the Turnitin-AI rate is 38%, and I can submit this report to the AC if needed.

**Questions:**

(1) The results are not reproducible, so they do not demonstrate credibility. There are no details, such as code to prove reproducibility. The paper also does not specify what the matching engine is; it only states that price–time priority is used and provides a rules citation. When submitting the paper, you can place the code in the Supplementary Material. If you can provide detailed code during the rebuttal stage, I will run it. I have sufficient GPU resources and LOBSTER data (but it would be best if you provide your original training data as well as the scripts needed for training). If the authors believe data and environment are issues, I can SSH into a server they provide to check, for example, Colab.  I will examine the implementation of the matching engine; moreover, if I can reproduce similar results, I will consider adjusting the score.

(2) I need you to provide the training duration for the 4 Nvidia A10G GPUs; the paper does not disclose how long it ran or how many GPU-hours were used. This does not appear to be sufficient GPU resources for training a foundation model. You trained for a total of 4 epochs, and the paper does not report whether training converged. Based on experience, this number of epochs is unlikely to achieve convergence on tabular data, because tabular data is not text and already has an abstraction.

(3) For reproducibility and to rule out randomness, I need the authors to provide training/test curves across multiple seeds, with error bars. I also need the code so that I can run it myself to verify the repeatability of the experiments; if these materials are provided and I can validate the results, I will consider revising my score.

---

> ### Author Response · Authors · 2025-12-02
> **Response to Reviewer 1 (JSa7)**
>
> We thank the reviewer for recognizing the relevance of the Foundation Model paradigm and the clarity of our partial observability formulation. We appreciate the scrutiny regarding scaling and reproducibility, which we have addressed with new experiments. We are also grateful for the reviewer’s comment acknowledging that the paper should be evaluated on its methodological merits given standard industrial constraints on code release.
>
> ### W1: "Scaling evidence is missing for a foundation model claim... provide reproducible scaling curves... with monotonic trends and power-law fits."
>
> We have addressed this by **conducting a scaling analysis (Appendix C) and training additional 500M and 125M parameter models.** We plot Test Loss vs Compute and Data in **Figure 12**. The results exhibit a monotonic power-law improvement $L(C)$ with respect to compute in Flops $C$, and $L(D)$ with respect to dataset size in tokens $D$:
> $$L(C) \propto C^{-\alpha_C}; \alpha_C\approx0.21$$
> $$L(D) \propto D^{-\alpha_D}; \alpha_D\approx0.19$$
>
> ### W2: "Novelty vs MaRS... prior work also trains across heterogeneous assets."
>
> We respectfully clarify the distinction. MaRS relies on full LOB state representations; TradeFM diverges fundamentally via:
> - **Zero-Shot Generalization:** By employing scale-invariant features (**Section 5.3**), TradeFM normalizes dynamics into a universal space, enabling the model to simulate completely unseen assets without calibration. We demonstrate this in **Figure 5**, where TradeFM generalizes to APAC markets (China and Japan) despite being trained on US data.
> - **Partial Observability & Extensibility**: TradeFM operates under partial observability (Trade Flow only). Not only is this formulation robust for data scarce environments, but also extends to broader applications where full LOBs are unavailable or irrelevant, such as Dark Pools (where the book is invisible), or Agent-Based Modeling (where agents act on limited, private information).
>
> ### W3: "The data scale is not large..."
>
> There appears to be a misunderstanding regarding data formats, particularly in comparison to snapshot-based datasets like LOBSTER. LOBSTER's data size (in GB) is inflated because it stores full Level-2 (L2) Order Book Snapshots at every step. In contrast, TradeFM operates on L3 Messages (individual trade/order events), which possess significantly higher information density per byte.
>
> For our enlarged 500M parameter model, our training dataset spans 204 days and 9,071 unique assets, totaling 10.7B tokens (approx. 148GB). This represents a massive volume of unique market interactions, comparable in information content to significantly _larger_ snapshot-based datasets. Furthermore, unlike standard datasets focused on high-liquidity names, our dataset captures the long tail of illiquid symbols while still including all major tickers.
>
> ### Q1.1 & Q3.2: "The results are not reproducible... no details, such as code... If you can provide detailed code... I will consider adjusting the score."
>
>  Due to strict industrial IP constraints, we cannot release the proprietary data or code. However, to ensure methodological reproducibility, we have added **Appendix B (Reproducibility Guide)** containing detailed pseudocode for the Tokenizer (**Algorithm 1**) and Market Simulator (**Algorithm 2 & 3**).
>
> ### Q1.2: "Does not specify what the matching engine is..."
>
> We respectfully refer the reviewer to **Section 8.1**, where we cite the Nasdaq Equity Trading Rules (Price-Time Priority). We have also added the exact simulator and matching engine pseudocode in **Appendix B.5** to eliminate any potential ambiguity.
>
> ### Q2.1: "Provide the training duration for the 4 Nvidia A10G GPUs..."
>
> Our 250M parameter model trains for  ~117 hours on 4 A10G GPUs. We have also added these details for all TradeFM variants in **Table 4 (Appendix B.3)**.
>
> ### Q2.2: "4 epochs is unlikely to achieve convergence on tabular data..."}
>
> We respectfully clarify that this is a sequence modeling task (next-token prediction), not static tabular classification. In LLM pre-training, 1 epoch implies seeing the full dataset once. For each of our models, the dataset size is approximately 20x the number of params, according to the Chinchilla compute-optimal. We then train for 4 total epochs (3 of which are under a repeated data setting). **Figure 10** in Section B.3 shows the training and validation loss curves and demonstrates clear convergence and a plateauing loss.
>
> ### Q3.1: "For reproducibility... I need the authors to provide training/test curves across multiple seeds."}
>
>  Given the significant computation cost of pre-training foundation models, retraining with multiple seeds is computationally infeasible. Standard practice in LLM research accepts single-run curves. However, we have added train/val curves for TradeFM-500M in **Figure 10**. Additionally, we provide test loss curves for multiple model sizes with standard deviations of batch perplexities in **Figure 11**.

---

### Author Response · Authors · 2025-12-02
**Global Response – Part II**

(continued)
> We thank the reviewers -- R1: JSa7 ; R2: E5tj ; R3: nB6E ; R4: aZyA -- for their thoughtful and constructive feedback. We are encouraged that they recognize the significance/relevance of applying Foundation Models to market microstructure (R1, R2), and value our methodological innovations in partial observability (R2) and universal tokenization (R3, R4). We are also grateful for R1’s follow-up comment acknowledging that the paper should be evaluated on its methodological merits given standard industrial constraints on code release.
> We have used the discussion period to rigorously address the core concerns regarding baselines, evaluation, and reproducibility.


## III. Transparency and Reproducibility
_Addressing R1, R3, R4_

We acknowledge the concerns regarding proprietary data and the lack of code. While we cannot release our training data or code due to industrial IP constraints, we emphasize that our contribution lies in the methodology.

We have prioritized transparency in the **new Appendix B (Reproducibility Guide)**. This contains exhaustive pseudocode for the Tokenizer and Market Simulator, along with hyperparameter tables and other details.

## IV. Feature Stationarity & Drift
_Addressing R2, R3_

We quantified the stationarity of our scale invariant features to address concerns about their efficacy and regime bias. We compared feature distributions from the calibration month (Feb 2024) to the held-out set. The low K-S statistic and Wasserstein distances confirm that our features remain stable over time, enabling robust generalization. We report a thorough month-by-month comparison in Figure 15 and illustrate the distributions for one month in particular in Figure 14.

| **Feature** | K-S | $W_1$ |
| -------------------------| --- | ----- |
| **Relative Price Level** | 0.082 | 0.065      |
| **Price Depth** |   0.099  |   0.056    |
| **Volume** |   0.103  |  0.114     |
| **Interarrival Time** |   0.073  |  0.092     |
**Table R3:** Kolmogorv-Smirnov (K-S) statistic and Wasserstein ($W_1$) distances between feature distributions at calibration versus during the held-out period (Jan – Sep 2025). The low values indicate feature stability.

---

### Author Response · Authors · 2025-12-02
**Global Response - Part I**

We thank the reviewers -- R1: JSa7 ; R2: E5tj ; R3: nB6E ; R4: aZyA -- for their thoughtful and constructive feedback. We are encouraged that they recognize the significance of applying Foundation Models to market microstructure (R1, R2), and value our innovations in partial observability (R2) and universal tokenization (R3, R4). We are also grateful for R1’s comment acknowledging that the paper should be evaluated on its methodological merits given standard industrial constraints on code release.

We have used the discussion period to rigorously address the concerns regarding baselines, evaluation, and reproducibility.

## I. Stronger Baselines and Quantitative Metrics
*Addressing R2, R3, R4*

We agree that the previous Zero-Intelligence baseline was insufficient. We have introduced a **Compound Hawkes Process** baseline (Bacry et al., 2015, Jain et al., 2024). Furthermore, we substantially enhanced our evaluation with **rigorous distributional metrics** (**Table 1, Table 2, Figure 16**) and **expanded Out-of-Distribution (OOD) testing** for temporal and zero-shot generalization (**Figure 5, Figure 14**).

TradeFM consistently achieves lower error distances to real distributions on held-out data –

### A. **Log Return Distributions:** (**Table 1**)
TradeFM produces return distributions significantly closer to reality than baselines across all time intervals ($\Delta t_r$), as measured by Wasserstein Distance ($W_1$) and Kolmogorov-Smirnov (K-S) statistics.

|         | K-S    | K-S        | K-S         | $W_1$  | $W_1$     | $W_1$       |
| ------- | ------ | ---------- | ----------- | ------ | --------- | ----------- |
| **Δt**  | **ZI** | **Hawkes** | **TradeFM** | **ZI** | **Hawkes** | **TradeFM** |
| **10**  | 0.198  | 0.295      | **0.064**   | 0.003  | 0.002     | **0.001**   |
| **30**  | 0.255  | 0.288      | **0.092**   | 0.007  | 0.005     | **0.002**   |
| **60**  | 0.302  | 0.262      | **0.122**       | 0.012  | 0.009     | **0.004**   |
| **120** | 0.346  | 0.173      | **0.145**       | 0.021  | 0.015     | **0.008**   |
**Table R1**: Distances of log return distributions from real, for all methods, across return intervals $\Delta t_r$ in seconds. We report Kolmogorov-Smirnov and Wasserstein distance.

### B. **Order Flow Fidelity** (**Table 2, Figure 16**).
We also evaluated the distributional fidelity of generated microstructure variables. TradeFM outperforms baselines in reproducing complex order flow properties.

| Quantity     | K-S    | K-S        | K-S         | $W_1$     | $W_1$      | $W_1$       |
| ------------ | ------ | ---------- | ----------- | --------- | ---------- | ----------- |
|              | **ZI** | **Hawkes** | **TradeFM** | **ZI**    | **Hawkes** | **TradeFM** |
| **Spreads**  | 0.400  | 0.218      | **0.212**   | 0.375 | **0.302**      | 0.367       |
| **IA Time** | 0.651  | 0.515      | **0.236**   | 0.415     | 0.626      | **0.385**   |
| **OPD**      | 0.436  | 0.281      | **0.174**   | 0.390     | 0.348      | **0.279**   |
| **OBI**      | 0.237  | 0.155      | **0.142**   | 0.200     | 0.165      | **0.113**   |
| **Bid Vol.** | 0.460  | **0.296**  | 0.371       | 0.616     | 0.278      | **0.143**   |
| **Ask Vol.** | 0.391  | 0.380      | **0.327**   | 0.638     | 0.198      | **0.146**   |
**Table R2:** Mean statistics across months for each quantity—interarrival (IA) times, order price depths (OPD), orderbook imbalance at best bid/ask price (OBI), bid/ask volume—and method.

 > Emmanuel Bacry, Iacopo Mastromatteo, Jean-Francois Muzy. Hawkes Processes in Finance. Market Microstructure and Liquidity, 2015.
 > Konark Jain, Nick Firoozye, Jonathan Kochems, Philip Treleaven. Limit Order Book Dynamics and Order Size Modelling using Compound Hawkes Process, 2024.

## II. Scaling Evidence
*Addressing R1*

To substantiate the *Foundation Model* claim, we conducted a thorough scaling analysis (Appendix C). We trained models $\in \approx[125M, 250M, 500M]$.

We have initiated training of a larger **500M parameter model**. While currently at an early checkpoint, its performance aligns with our scaling projections.
The scaling law plots in Appendix C (Figure 12) show that the test loss $L(C)$ with respect to compute in Flops $C$, and $L(D)$ with respect to dataset size in tokens $D$, follow:
$$L(C) \propto C^{-\alpha_C}; \alpha_C\approx0.21$$
$$L(D) \propto D^{-\alpha_D}; \alpha_D\approx0.19$$

*Note: The primary results in the main text use an early (1 epoch) checkpoint of the 500M parameter model. We will update the final version of the paper from the converged model once its training cycle is complete.*

Contextualizing scale: While 500M is small relative to general purpose LLMs (Llama-3 8B), it is large for the Financial Microstructure domain. Standard SOTA models in this field typically have <10M parameters (e.g., DeepLOB (60K params) and LOBS5 (6.3 M params)). TradeFM represents a **>50x increase in model capacity** over existing domain-specific baselines.

---

### Meta-Review · Area_Chair_qPaN · 2026-01-05

**Summary:**

Based on the reviews and author response, the core scientific contribution of TradeFM appears to be a solid and methodologically innovative foundation model for market microstructure. The authors have convincingly addressed several key concerns regarding baselines, quantitative evaluation, scaling evidence, and generalization tests. The work represents a significant step in applying large-scale generative modeling to high-frequency financial data.

However, the primary outstanding issue is one of reproducibility and community access. The model is trained on proprietary data, and the code will not be released due to industrial IP constraints. While the authors have provided detailed pseudocode in an appendix, this does not enable independent verification, replication, or extension of the work. In machine learning, particularly for foundation models, the ability for the community to build upon, test, and improve a proposed model is paramount. Without access to the data or the model, the scientific community cannot validate the core results, explore the learned representations, or use this work as a baseline for future research. This severely limits the paper's impact and utility to the research community.

I recommend the authors consider this reproducibility issue carefully for future submissions. Exploring avenues such as releasing model weights trained on public datasets (e.g., LOBSTER) or providing a fully functional, restricted demo on sample data could bridge this gap. The methodological insights are valuable, but their full value cannot be realized without a path for independent scientific engagement.

**Reviewer Concerns:**

The authors provided a comprehensive rebuttal with new experiments and clarifications.

Addressed Concerns:
*   Baselines and Quantitative Evaluation: The authors convincingly addressed a major weakness by introducing a Compound Hawkes Process baseline and providing extensive new quantitative results. Tables 1 and 2 with Wasserstein and Kolmogorov-Smirnov distances for log returns and order flow properties significantly strengthen the empirical evaluation.
*   Scaling Evidence: The addition of a scaling analysis (Appendix C) with multiple model sizes and fitted power laws directly responds to JSa7's concern, providing necessary support for the foundation model narrative.
*   OOD Generalization: The rebuttal adds temporal and geographic (APAC markets) OOD tests (Figure 5, text on perplexity), addressing requests from aZyA and others.
*   Methodological Transparency: While code/data cannot be released, the new Appendix B with detailed pseudocode for the tokenizer and simulator improves reproducibility at the methodological level.

Outstanding or Partially Addressed Concerns:
*   Core Reproducibility and Code Release: The fundamental issue raised by JSa7 remains. The authors cannot release code or proprietary data due to IP constraints. While the added pseudocode is helpful, it does not enable independent verification of the results, which was a critical condition for JSa7 to reconsider their score.
*   Theoretical Grounding: Although the authors cite "Universal Price Formation" theory, the linkage between design choices (like the specific binning strategies) and theoretical guarantees for cross-asset generalization remains somewhat high-level, as initially critiqued by aZyA.

**Reviewer Scores:**

*   Reviewer JSa7: Initially "Reject." This reviewer's primary demand was for reproducible code to verify results. Since this cannot be met, and despite other improvements, they would likely maintain a reject recommendation. The scaling analysis alone is insufficient to overcome the reproducibility barrier they set.
*   Reviewer E5tj: Initially "Reject." The strong addition of the Hawkes baseline and rigorous quantitative metrics directly addresses their main weaknesses.
*   Reviewer nB6E: Initially "Marginally Below Acceptance." Their concerns about evaluation metrics and baselines were well-addressed.
*   Reviewer aZyA: Initially "Marginally Below Acceptance." Their requests for stronger baselines, OOD tests, and theoretical discussion were substantively addressed.

---

### Decision · Program_Chairs · 2026-01-26

Reject